# Potential-Vorticity Dynamics of Troughs and Ridges within Rossby Wave Packets during a 40-year reanalysis period

Franziska Teubler and Michael Riemer

Institute for Atmospheric Physics, Johannes Gutenberg-Universität Mainz, Mainz, Germany

**Correspondence:** Franziska Teubler (f.teubler@uni-mainz.de)

**Abstract.** Rossby wave packets (RWPs) are fundamental to midlatitude dynamics and govern weather systems from their individual life cycles to their climatological distributions. Renewed interest in RWPs as precursors to high-impact weather events and in the context of atmospheric predictability motivates this study to revisit the dynamics of RWPs. A quantitative potential vorticity (PV) framework is employed. Based on the well established PV-thinking of midlatitude dynamics, the processes governing RWP amplitude evolution comprise group propagation of Rossby waves, baroclinic interaction, the impact of upper-tropospheric divergent flow, and direct diabatic PV modification by nonconservative processes. An advantage of the PV framework is that the impact of moist processes is more directly diagnosed than in alternative, established frameworks for RWP dynamics. The mean dynamics of more than 6000 RWPs from 1979-2017 are presented using ERA5 data, complemented with nonconservative tendencies from the ‚Year of tropical convection' data (available 2008-2010).

Confirming a pre-existing model of RWP dynamics, group propagation within RWPs is consistent with linear barotropic theory, and baroclinic and divergent amplification occur most prominently during the mature stage and rather towards the trailing edge of RWPs. Refining the pre-existing model, the maximum of divergent amplification occurs in advance of maximum baroclinic growth and baroclinic interaction tends to weaken RWP amplitude towards the leading edge. ‚Downstream baroclinic development' is confirmed to provide a valid description of RWP dynamics in both, summer and winter, although baroclinic growth is substantially smaller (about 50%) in summer. Longwave radiative cooling makes a first-order contribution to ridge and trough amplitude, with the potential that this contribution is partly associated with cloud-radiative effects. The direct impact of other nonconservative tendencies, including latent heat release, is an order of magnitude smaller than longwave radiative cooling. Arguably, latent heat release still has a substantial impact on RWPs by invigorating upper-troposhperic divergence. The divergent flow amplifies ridges and weakens troughs. This impact is of leading order and larger than that of baroclinic growth. To the extent that divergence is associated with latent heat release below, our results show that moist processes contribute to the well-known asymmetry in the spatial scale of troughs and ridges. For ridges, divergent amplification is strongly coupled to baroclinic growth and enhanced latent heat release. We thus propose that the life cycle of ridges is best described in terms of ‚downstream *moist*-baroclinic development'. Consistent with theories of moist-baroclinic instability, both the amplitude and the relative location of latent heat release within the developing wave pattern depends on the state of the baroclinic development. Taking this "phasing" aspect into account, we provide some evidence that variability in the strength of divergent ridge amplification can predominantly be attributed to variability in latent heat release below, rather than to secondary circulations associated with the dry dynamics of a baroclinic wave.

# 1 Introduction

Rossby wave packets (RWPs) propagating along the midlatitude jet (Wirth et al., 2018) are fundamental to both, the individual
evolution and the climatological distribution of midlatitude weather systems. Specifically, RWPs organize the formation, inten-
sification, and movement of weather systems and special attention has been given to RWPs as precursors to extreme weather
events (e.g., Shapiro and Thorpe, 2004; Martius et al., 2008; Wirth and Eichhorn, 2014; Piaget et al., 2015; Grazzini et al.,
2020). Beyond individual weather systems, the recurrent occurrence of RWPs has been associated with periods of temperature
extremes (Röthlisberger et al., 2019). And from a climatological perspective, the dynamics of midlatitude storm tracks can be
described in terms of the excitation, propagation, and decay of RWPs (Chang and Orlanski, 1993; Chang et al., 2002).

A defining characteristic of RWPs is the associated downstream dispersion of energy. This downstream dispersion implies
that RWPs connect the current evolution of weather systems with the previous evolution of weather systems in the upstream
region, i.e., RWPs provide a means of teleconnection between systems. The significance of this characteristic for atmospheric
predictability has long been recognized (Cressman, 1948; Hovmöller, 1949). Smaller-scale weather features embedded in
RWPs may inherit some of this putative predictability (Anthes et al., 1985; Grazzini, 2007; Grazzini and Vitart, 2015). On the
other hand, however, forecast errors and uncertainty originating from weather systems upstream may propagate within RWPs
and may severely compromise predictability in the downstream region (e.g., Anwender et al., 2008; Rodwell et al., 2013). In
fact, midlatitude forecast errors have been shown to grow and maximize within RWPs (Dirren et al., 2003; Davies and Didone,
2013; Baumgart et al., 2018) and the reliable medium-range prediction of RWPs constitutes a challenge for state-of-the-art
numerical forecast systems (Glatt and Wirth, 2014; Gray et al., 2014).

The renewed interest in RWPs due to their role in atmospheric extremes and predictability provides motivation to revisit the
dynamics governing RWP evolution. The prevailing paradigm for RWP dynamics, developed based on RWPs during winter,
has been dubbed downstream baroclinic development (e.g., Orlanski and Sheldon, 1995; Chang, 2000). In this paradigm,
development at the leading edge of the wave packet is governed by downstream dispersion of wave energy, consistent with
linear (barotropic) RWP dynamics. Subsequently, the developing perturbation further grows by baroclinic energy conversion.
The decay of the perturbation at the mature stage is then initiated by downstream dispersion of energy and the cycle may repeat
itself farther downstream. Basically, the paradigm of downstream baroclinic development describes the baroclinic coupling of
RWPs. The paradigm, however, does not explicitly consider moist processes. In general, moist processes increase baroclinic
growth by associated latent heat release and effectively reducing static stability (Emanuel et al., 1987; Gutowski et al., 1992)
and by interactions between the diabatically generated PV anomalies and boundary theta anomalies (Mak, 1982; de Vries
et al., 2010). More recent studies strongly indicate that the impact of moisture differs substantially between ridges and troughs.
Many studies have demonstrated significant ridge amplification by latent heat release below and argue that associated upper-
tropospheric divergent outflow plays a crucial role in this amplification (e.g. Davis et al., 1996; Riemer et al., 2008; Grams et al.,
2011; Archambault et al., 2013; Pfahl et al., 2015; Grams and Archambault, 2016; Steinfeld and Pfahl, 2019). A large case-to-
case variability between individual ridges, however, can be expected (Teubler and Riemer, 2016). The impact on troughs, in
contrast, is less extensively studied and the few existing studies indicate a more complex impact of upper-tropospheric outflow

on troughs and a potentially detrimental impact on trough amplitude (Pantillon et al., 2013; Riemer and Jones, 2014; Teubler and Riemer, 2016).

Moist processes are of particular interest in the context of atmospheric predictability. Forecast errors grow most rapidly in regions of convection and precipitation (Hohenegger and Schär, 2007; Zhang et al., 2007; Selz and Craig, 2015). Moist processes in the warm sector of cyclones have been identified as one of the most important sources of forecast errors and uncertainty in the midlatitudes (Rodwell et al., 2018; Sanchez et al., 2020). Upper-tropospheric outflow most effectively communicates uncertainties associated with moist processes to the tropopause region (Baumgart et al., 2019; Baumgart and Riemer, 2019), where these uncertainties have been shown to potentially transfer to the amplitude of the downstream ridge (e.g. Martínez-Alvarado et al., 2016; Grams et al., 2018) and thus eventually RWP amplitude (Baumgart et al., 2019; Ghinassi et al., 2020). Understanding the predictability of RWPs as large-scale atmospheric features thus requires understanding of the occurrence and characteristics of moist processes within RWPs.

This study revisits the dynamics of ridges and troughs within RWPs in a quantitative potential-vorticity (PV) framework. For the first time, a quantitative analysis of RWP dynamics will be performed for a very large number (over 6000) and year-round occurrence of cases in the northern hemisphere. The PV framework has been developed in Teubler and Riemer (2016), building on previous work by Davis and Emanuel (1991), Nielsen-Gammon and Lefevre (1996), and Riemer et al. (2008). The framework has previously been employed in case studies (Piaget et al., 2015; Teubler and Riemer, 2016; Schneidereit et al., 2017), a climatological study of extreme precipitation events (Grazzini et al., 2020), and to investigate the PV dynamics of forecast errors and ensemble-forecast spread (Baumgart et al., 2018, 2019; Baumgart and Riemer, 2019). Essentially, the framework constitutes a quantification of the well-established PV thinking of midlatitude dynamics (e.g., Hoskins et al., 1985). PV thinking provides dynamical understanding by considering the evolution and interaction of PV anomalies, which maximize at lower and upper levels in the midlatitude troposphere. A conceptual separation into two layers thus captures the essence of Rossby wave propagation and baroclinic development (Eady, 1949; Phillips, 1951; Heifetz et al., 2004a, b)[1].

A comprehensive description of the PV perspective on RWP dynamics can be found in Wirth et al. (2018, their Sect. 3f). An alternating succession of positive and negative synoptic-scale, upper-level PV anomalies constitute an RWP. Consideration of these upper-level PV anomalies and a background PV gradient in isolation describes (quasi-barotropic) Rossby wave dynamics: Upper-level PV advection by the winds associated with the upper-level PV anomalies themselves signify intrinsic phase and group propagation. The impact of low-level PV anomalies describes baroclinic interaction: Upper-level PV advection by the winds associated with these low-level anomalies signify baroclinic growth (or weakening). This strictly balanced conceptual model can be complemented by including PV tendencies due to advection by divergent flow, which is not included in the balanced flow (under nonlinear balance), and due to nonconservative processes. With respect to nonconservative processes, we follow the convention used by Davis et al. (1993): Nonconservative PV tendencies are referred to as direct nonconservative impact. Advective tendencies associated with nonconservative processes are referred to as indirect nonconservative impact. One prominent indirect nonconservative impact are advective tendencies by the winds associated with low-level PV anomalies

[1]The concept can be extended to include mid-level PV anomalies and humidity (de Vries et al., 2009, 2010). This extension, however, will not be considered in the current study.

generated by latent heat release, in particular their role in enhancing baroclinic growth. This impact, however, will not be given special attention in the current study. Instead, this study focuses on PV advection by the divergent flow invigorated by latent heat release below as a prominent indirect nonconservative impact. It is still an open question, however, to what extent upper-tropospheric divergence is associated with moist and dry (balanced) dynamics, respectively.

The PV perspective provides a diagnostic framework that is complementary to the often-used eddy kinetic energy framework (e.g., Orlanski and Sheldon, 1995; Chang, 2000; Chang et al., 2002). Both frameworks have their strengths and weaknesses and a detailed comparisons of the two frameworks can be found in Teubler and Riemer (2016, their Sect. 3f) and Wirth et al. (2018, their Sect. 3f). A notable caveat of our PV framework is that the effect of deformation on the evolution of PV anomalies is not accounted for. Deformation is of particular importance during wave breaking and the associated decay of PV anomalies. This important caveat needs to be borne in mind when interpreting the results of the PV analysis in the late stage of the life cycle of individual troughs and ridges. Arguably, the most substantial advantage of the PV framework is that the impact of nonconservative processes is much more directly diagnosed than in an eddy kinetic energy framework. One focus of this study is on the impact of moist processes on RWPs and we thus adopt the PV perspective.

The overarching goal of this study is to provide a robust mean-picture of the dynamics of troughs and ridges in real-world RWPs. This mean evolution may provide a benchmark for subsequent studies to identify anomalous dynamical behaviour in more specific scenarios. Main questions to be addressed in the current study are:

- To what extent does the paradigm of downstream baroclinic development provide a useful description of RWP evolution during summer, when baroclinic coupling is relatively weak?

- How do nonconservative processes modify the paradigm of downstream baroclinic development?

- What is the relative role of direct diabatic PV modification and the indirect impact of latent heat release by invigorating upper-tropospheric divergence?

- And to what extent can upper-tropospheric divergence be attributed to moist and dry (balanced) dynamics, respectively?

The next section describes the data and introduces the quantitative PV framework. Section 3 explains how RWPs are selected from a pre-existing catalogue and how their associated troughs and ridges are identified. In addition, an account of the accuracy of our PV diagnostic is given. The subsequent two sections present our results: Section 4 considers spatial patterns of PV anomalies, piecewise (advective) PV tendencies, and a proxy for latent heat release, whereas Sect. 5 focuses on the temporal evolution and relation of individual processes. Our conclusions and a final discussion are given in Sect. 6.

## 2 Data and quantitative PV framework

### 2.1 Data

This study uses two different data sets: i) the "Year of tropical convection" (YOTC) data (e.g., Moncrieff et al., 2012) based on the integrated forecast system of the European Centre for Medium-Range Weather Forecasts (ECMWF) and ii) the ECMWF

re-analysis ERA5 (Hersbach et al., 2019). The YOTC data is unique in the sense that it contains model tendencies from the different physical parameterization schemes. These tendencies are computed from 36-h forecasts starting daily from the 1200 UTC analysis and are accumulated over 3 h. The YOTC data is available from May 2008 to April 2010 every 6 h and has previously been used in case studies to quantify the direct impact of nonconservative processes on trough and ridge dynamics (Teubler and Riemer, 2016; Schneidereit et al., 2017). ERA5 data is publicly available since 1979 and we use the data from June 1979 to November 2017 every 3 h. A 3-hourly resolution is deemed sufficient to analyze the impact of organized moist processes (e.g., recurving tropical cyclones, warm conveyor belts, and mesoscale convective systems) on RWPs. We therefore did not exploit the available hourly resolution of the ERA5 data to avoid excessively large data handling and computational cost. For both, YOTC and ERA5, we use a spatial resolution of 1° and 17 pressure levels (1000, 950, 925, 900, 850, 800, 700, 600, 500, 400, 300, 250, 200, 150, 100, 70, and 50 hPa), from which data is interpolated to a 50 hPa vertical resolution by cubic spline interpolation.

## 2.2 Quantification of individual processes: piecewise PV tendency framework

The quantitative piecewise PV tendency framework employed in this study has been introduced by Teubler and Riemer (2016). The framework considers Ertel's PV (Ertel, 1942) in its hyodrostatic approximation on isentropic levels:

$$PV = \frac{(\zeta_\theta + f)}{\sigma}, \tag{1}$$

where $\zeta_\theta$ is the component of relative vorticity perpendicular to an isentropic surface, $f$ the Coriolis parameter and $\sigma = -g^{-1}(\partial p/\partial \theta)$ the isentropic layer density with gravity $g$, pressure $p$, and potential temperature $\theta$.

The PV tendency equation, neglecting small nonhydrostatic effects, is given by isentropic advection and nonconservative PV modification ($\mathcal{N}$):

$$\frac{\partial PV}{\partial t}\bigg|_\theta = -\mathbf{v} \cdot \nabla_\theta PV + \mathcal{N}, \tag{2}$$

with $\mathbf{v} = (u, v, 0)$ the wind vector and $\nabla_\theta = (\partial_x, \partial_y, \partial_\theta)$ the gradient operator along an isentropic surface. The nonconservative PV modification (cmp. Hoskins et al., 1985)[2] is given by

$$\mathcal{N} = -\dot{\theta}\frac{\partial PV}{\partial \theta} + \frac{1}{\sigma}\left[(\boldsymbol{\nabla}_\theta \times \mathbf{v} + f\mathbf{k}) \cdot \boldsymbol{\nabla}_\theta \dot{\theta} + \mathbf{k} \cdot (\boldsymbol{\nabla}_\theta \times \dot{\mathbf{v}})\right], \tag{3}$$

with $\mathbf{k} = (0, 0, 1)$ the unit vector perpendicular to an isentropic surface. To evaluate the tendency equation, the nonconservative heating rate $\dot{\theta}$ and the nonconservative momentum sources and sinks $\dot{\mathbf{v}}$ are calculated from the 3-hourly accumulated YOTC tendencies as forward finite differences. The nonconservative tendencies available in the YOTC data are those from the parameterization schemes of longwave radiation, convection, clouds, and turbulence and orographic drag (ECMWF, 2009). By evaluating Equation 2, a residual occurs. Using YOTC data this residual comprises the (small) missing tendencies from shortwave radiation, nonconservative effects of the dynamical core (numerical diffusion), model deficiencies due to the parameterizations, and numerical inaccuracies associated with using a finite time step. For ERA5, nonconservative tendencies from

---

[2]Note, there is a wrong sign in equation (72) and (73) in Hoskins et al. (1985).

all individual parameterizations, in particular from the cloud and the convection scheme, are not available and we thus evaluate the isentropic advective tendencies in Equation 2 only.

The advection term in the PV tendency equation (Equation 2) is further partitioned to quantitatively represent the PV perspective of midlatitude dynamics as described in the introduction. This partitioning is applied to the horizontal wind on pressure levels, from which the individual wind components are interpolated to isentropic levels to calculate the associated (piecewise) advective PV tendencies[3]. First, a Helmholtz partitioning is applied to decompose the flow into its irrotational and nondivergent components (following version 5 in Lynch, 1989). The resulting harmonic component is negligible and thus we will refer hereafter to the irrotational wind as divergent wind ($\mathbf{v}_{div}$). Then, the nondivergent component is further decomposed by piecewise PV inversion based on nonlinear balance (Charney, 1955; Davis and Emanuel, 1991; Davis, 1992). PV anomalies ($PV'$) are defined as deviations from a 30-day mean background state $\overline{PV}$. The 30-day period is centered on the respective life time of each RWP, i.e., a constant background state is used for each RWP. A 30-day period has been chosen because it is long enough to be considered as steady in the sense that $\overline{\mathbf{v}} \cdot \boldsymbol{\nabla}\overline{PV} \ll \mathbf{v}' \cdot \boldsymbol{\nabla}\overline{PV}$, where $\overline{\mathbf{v}}$ is the background wind and $\mathbf{v}'$ the wind associated with the respective PV anomalies, and short enough that the associated anomalies can be considered to be synoptic-scale features. According to Bretherton (1966), $\theta$ anomalies on the upper and lower boundary of a domain can be interpreted as PV anomalies also.

PV anomalies are partitioned into upper-level and lower-level PV anomalies. Following previous work (Davis et al., 1996; Riemer and Jones, 2010, 2014; Teubler and Riemer, 2016) the separation level between upper- and lower-level PV anomalies is chosen to be between 600 and 650 hPa. In general, midlatitude PV anomalies at such a mid-tropospheric level are small compared to lower- and upper-tropospheric anomalies; exceptions comprise deep tropopause folds (e.g. Donnadille et al., 2001), recurving tropical cyclones (e.g. Thorncroft and Jones, 2000), and midlatitude cyclones with deep-tropospheric PV towers (e.g. Rossa et al., 2000). The occurrence of these exceptions, however, is infrequent and we are thus confident that they do not affect the statistics presented in this study.

The upper and lower boundary $\theta$ anomalies are included in our definition of the upper- and lower-level PV anomalies, respectively. The lower boundary $\theta$ anomalies include the thermal anomalies of baroclinic waves, e.g., warm and cold sectors of cyclones, and a contribution from the static stability anomalies associated with low-level (interior) PV anomalies. Our upper boundary is situated in the lower-most stratosphere intersecting PV anomalies that are associated with RWPs. The upper boundary $\theta$ anomalies thus include the static stability anomalies associated with these PV anomalies. In addition, the boundary $\theta$ anomalies may include contributions from distant PV anomalies. Namely, the lower boundary $\theta$ anomaly may include contributions from PV anomalies above the separation level and the upper boundary $\theta$ anomaly contributions from PV anomalies below the separation level and from stratospheric PV anomalies outside of our domain that are unrelated to RWPs. These contributions are ultimately due to vertical motion associated with the evolution of the distant anomalies (cf.

---

[3]This procedure neglects adiabatic vertical motion in the transformation from pressure to isentropic levels. Adiabatic vertical motion can be expected to be 2-3 orders of magnitude smaller than the horizontal wind on pressure levels near the jet, where we evaluate the tendencies. The second author had explicitly verified in previous work that neglecting vertical motion on pressure levels when calculating PV tendencies on isentropes does not affect the diagnostic in any notable quantitative sense even during upper-level frontogenesis (as, e.g., in Riemer and Jones, 2010), when the slope of isentropes are a maximum.

the concept of a "very gentle 'vacuum cleaner'" in Sect. 4 of Hoskins et al. (1985)). Vertical motion at the upper and lower boundary, however, is strongly limited by the large static stability in the stratosphere and the closeness of the boundary to the rigid boundary of the Earth's surface, respectively. Contributions of distant PV anomalies can thus be expected to be negligibly small, and the interpretation of the boundary $\theta$ anomalies as upper-and lower-level PV anomalies appears to be very reasonable.

The piecewise PV inversion uses the so-called subtraction (ST) method proposed by Davis (1992). The inversion is performed on a horizontal domain that extends from 25°N - 80°N and that is periodic in longitude to reduce boundary effects. In the vertical, $\theta$ is specified as Neumann boundary condition at 875 hPa and 125 hPa. For consistency, $\theta$ is calculated from the pressure level data as $\theta = -\partial\phi/\partial\Pi$, with geopotential $\phi$ and the Exner function $\Pi = (p/p_0)^{R_d/c_p}$, where $p_0 = 1000$ hPa, $R_d$ is the gas constant and $c_p$ the heat capacity for dry air. The interior levels range from 850 hPa to 150 hPa, every 50 hPa. The wind field obtained by inversion is interpolated to isentropic levels using log-linear interpolation, i.e., linear interpolation under the assumption that temperature varies linearly with the natural logarithm of pressure. The isentropic levels suitable to analyze RWPs along the midlatitude jet are subject to a seasonal cycle. To account for the seasonal dependence, we follow the recommendations of Röthlisberger et al. (2018) and use the following isentropic levels: 320 K for December, January, February and March, 325 K for April and November, 330 K for May and October, 335 K for June and September, and 340 K for July and August. In the following, PV advection will refer to PV advection on these isentropic levels. In summary, our decomposition of the horizontal wind reads

$$\mathbf{v} = \mathbf{v}_{qb} + \mathbf{v}_{bc} + \mathbf{v}_{div} + \mathbf{v}_{res}. \tag{4}$$

The quasi-barotropic (near-tropopause) dynamics of the RWP is represented by PV advection due to $\mathbf{v}_{qb}$, which is defined as the sum of $\overline{\mathbf{v}}$ and the wind associated with the upper-level PV anomalies (including the upper-boundary $\theta$ anomalies). Baroclinic impact on RWP dynamics in the PV framework is represented by PV advection due to $\mathbf{v}_{bc}$, which is defined as the wind associated with the lower-level PV anomalies and is dominated by the lower-boundary $\theta$ anomalies (not shown). The impact of upper-tropospheric divergence is represented by $\mathbf{v}_{div}$. In addition we introduce the residual $\mathbf{v}_{res}$, which arises i) due to inherent features of piecewise PV inversion on a limited domain under nonlinear balance, namely imperfect knowledge of the boundary conditions and the nonlinearity of the balance condition, and ii) due to numerical inaccuracies, mostly in calculating the Neumann boundary condition at 125 hPa, where the vertical $\theta$ gradient is very large, and in the interpolation from pressure to isentropic levels. Figure 7 shows how these inherent and numerical inaccuracies of our piecewise PV inversion affect the results. On average, the relative uncertainty is small. For troughs during extended winter (Figure 7b), however, there is a persistent relative uncertainty of 25% - 30% in the baroclinic component, when the uncertainty is split equally between the baroclinic and the quasi-barotropic tendency. Despite this relatively large uncertainty, and even if all of the uncertainty were attributed to the baroclinic component, none of the results of our study would be qualitatively affected.

## 2.3 Amplitude evolution of troughs and ridges

This study focuses on the amplitude evolution of troughs and ridges. Individual troughs and ridges are defined in our framework as the positive PV anomaly in-between two ridge axes and the negative PV anomaly in-between two trough axes, respectively.

Trough and ridge axes are defined based on isolines of zero meridional wind anomaly (for details see chapter 3d in Teubler and Riemer, 2016). This simple identification works very well until the evolution becomes highly nonlinear and wave breaking occurs. We define the amplitude of troughs and ridges by the associated spatially-integrated PV anomaly ($\int_{\mathcal{A}(t)} PV' dA$). With this definition, the amplitude evolution [in PVU m$^2$/s] of troughs and ridges is given by (derivation can be found in Teubler and Riemer, 2016)

$$225 \quad \frac{d}{dt} \int_{\mathcal{A}(t)} PV' dA = \int_{\mathcal{A}} \left[ -\mathbf{v} \cdot \boldsymbol{\nabla} \overline{PV} - PV' (\boldsymbol{\nabla} \cdot \mathbf{v}) + \mathcal{N} \right] dA + \mathcal{B}nd, \tag{5}$$

where $\mathcal{A}$ denotes the area of the respective PV anomaly. Note, this measure quantifies the absolute and not the relative growth of anomalies to avoid an bias during the first stage of the anomaly evolution when anomalies are very small and hard to detect. Amplitude evolution is thus governed by four mechanisms: advection of background PV, divergence within the PV anomaly, nonconservative processes, and a boundary term $\mathcal{B}nd = \int_{\mathcal{S}} PV' (\mathbf{v} \cdot \mathbf{n}) \, dS$, which describes the net flux of PV anomalies

across the curve $\mathcal{S}$ (with normal vector $\mathbf{n}$) that defines area $\mathcal{A}$. The boundary term is usually small because $PV' = 0$ along most of the boundary $\mathcal{S}$ (see Figs. 4 and 5 in Teubler and Riemer, 2016). The boundary term may become large when our simple algorithm fails to correctly identify anomalies during their highly nonlinear evolution. We will use this characteristic below to restrict our analysis to troughs and ridges that evolve rather benignly.

     Using our decomposition of the wind (Equation 4) in the amplitude evolution (Equation 5) finally yields the individual

contributions that will be considered in the remainder of this study:

$$\frac{d}{dt} \int_{\mathcal{A}(t)} PV' \, dA = - \int_{\mathcal{A}} \mathbf{v}_{qb} \cdot \boldsymbol{\nabla} \overline{PV} \, dA - \int_{\mathcal{A}} \mathbf{v}_{bc} \cdot \boldsymbol{\nabla} \overline{PV} \, dA - \int_{\mathcal{A}} [\mathbf{v}_{div} \cdot \boldsymbol{\nabla} \overline{PV} + PV' (\boldsymbol{\nabla} \cdot \mathbf{v}_{div})] \, dA + \tag{6}$$

$$+ \int_{\mathcal{A}} \mathcal{N} \, dA + \mathcal{B}nd - \int_{\mathcal{A}} [\mathbf{v}_{res} \cdot \boldsymbol{\nabla} \overline{PV} + PV' (\boldsymbol{\nabla} \cdot \mathbf{v}_{res})] \, dA \, .$$

The winds obtained from PV inversion ($\mathbf{v}_{qb}$ and $\mathbf{v}_{bc}$) are nondivergent and thus contribute to the amplitude evolution only by advection of background PV (first two terms on the RHS), whereas the third term denotes the combined impact of the divergent wind.

wind. The last term quantifies the residual in the PV budget due to the residual in the wind decomposition.

## 3    Selection of PV anomalies for further analysis

### 3.1    Selection of RWPs and their associated troughs and ridges

We select our RWP cases from a pre-existing catalogue of RWPs (Wolf and Wirth, 2017). This catalogue is based on ERA-interim data (Dee et al., 2011). RWPs in this catalogue are identified by an object-based tracking algorithm with a 12-hourly

resolution. Identification of RWP objects is based on thresholding the envelope field of the 300 hPa meridional wind. The envelope is calculated along streamlines of a zonally varying background state following Zimin et al. (2006) with Wolf and Wirth (2017) using a latitude-dependent wave number filter before envelope calculation.

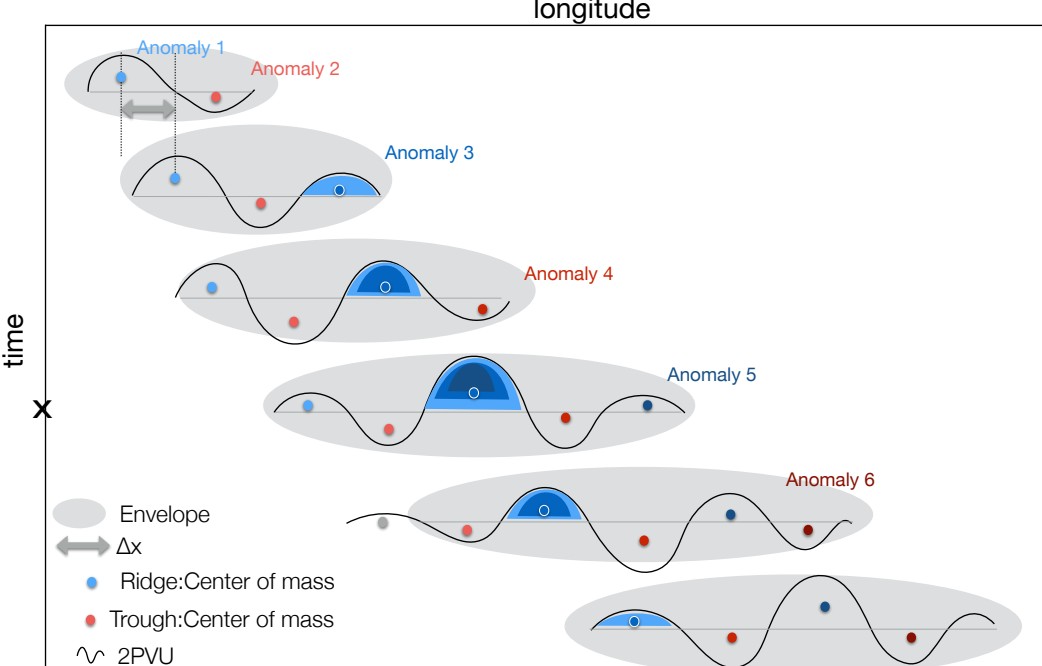

**Figure 1.** Schematic of envelopes (grey shading) and anomaly evolution within a RWP. The black line depicts the 2 PVU contour illustrating the tropopause. Red and blue dots represent the center of mass of negative (ridges) and positive (troughs) PV anomalies, respectively. The blue shading represents the amplitude evolution of Anomaly 3 with the time of maximum amplitude denoted by x along the time axis. For further details we refer to the text.

For the current study, we consider RWPs with a lifetime between 4 and 15 days. Individual troughs and ridges are defined to be part of a specific RWP if the zonal position of the center of mass of their respective PV anomaly is located within the zonal extent of the envelope field of that RWP object. To maximize the number of troughs and ridges identified within these RWPs, we have recalculated the envelope field with the YOTC and ERA5 data following Wolf and Wirth (2017) but with a lower envelope threshold of 15 m/s to enlarge RWP objects[4]. Individual troughs and ridges are tracked with time using a simple distance criterion: If the center of mass between two consecutive trough (ridge) anomalies is below a threshold $\Delta x = 650\,\mathrm{km}$, then the anomalies are considered to represent the same trough (ridge). Otherwise, a new trough (ridge) is identified as part of that RWP. The procedure is performed for each time step as long as the RWP object is identified. For consistency, we use $\Delta x/2$ for the 3-hourly ERA5 data. Our threshold distance $\Delta x$ is within 80 % of the smallest distances between consecutive ridges and within 84 % of the smallest distance between consecutive troughs. For a linear wave, our threshold distance implies a phase speed of $30\,\mathrm{m\,s^{-1}}$, which is rather high compared to typically observed values. Nonlinear effects however, mostly the deformation of the PV anomaly, may yield such relatively large zonal displacements of the center of mass of a PV anomaly and

---

[4]This threshold only affects the number of troughs and ridges identified but does not affect the results shown below

thus we here use such an inclusive criterion. Furthermore, neighboring troughs (ridges) are virtually always much farther apart than our threshold distance and thus erroneous matching of neighboring troughs (ridges) at consecutive times is not expected. The identification and tracking of individual trough and ridge anomalies is illustrated in Fig. 1.

### 3.2 Eliminating data of questionable representiveness

Before performing statistical analysis and creating composites of our data, we eliminate tendency data for which it is questionable that they represent well the evolution of anomalies within RWPs. Our elimination criteria refer to the identification and the life time of anomalies. First, we eliminate data from time steps at which the absolute value of the boundary term ($\mathcal{B}nd$) is exceptionally large. To define exceptionally large values we apply an often-used method to define outliers, the so-called interquartile range (IQR) rule. The $n$-IQR rule defines outliers as those values that lie $n$ times the IQR outside of the first and the third quartile. For the boundary term, we apply a standard choice of $n = 3$, i.e.,the 3-IQR rule. As noted above, exceptionally large values of the boundary term indicate that individual anomalies are not identified correctly as for example during highly nonlinear evolution. Arguably, the tendencies diagnosed for these anomalies do not represent the actual evolution. Second, we stop tracking the anomaly when the difference between the diagnosed and the observed amplitude evolution is exceptionally large (defined again by the 3-IQR rule). The observed amplitude evolution is defined as the forward finite difference between the amplitude at two consecutive time steps. Our interpretation of these exceptionally large differences is that splitting or merging of anomalies occurs, because these events substantially change the (spatially integrated) amplitude of the anomaly but are not captured by our diagnostic. After the assumed splitting or merging, respective anomalies are redefined as new, separate anomalies. Third, we eliminate data when this difference is large (defined by the 1.5-IQR rule; see next subsection for the choice of the threshold). Finally, we require that anomalies exist for at least 2 days. We consider shorter lived anomalies as not being representative of troughs and ridges that develop within RWPs. After eliminating data according to these criteria, 111 RWPs with 354 ridges and 321 troughs are considered during the YOTC period and 6311 RWPs with 15651 ridges and 16146 troughs during the ERA5 period.

### 3.3 Verification of the PV budget

Despite eliminating data of questionable representiveness, differences between the observed and the diagnosed amplitude tendencies may still occur due to uncertainties in the calculation of the boundary term $\mathcal{B}nd$, the residual wind component $\mathbf{v}_{res}$, the comparison of instantaneous tendencies with finite-time differences and mainly due to small-scale merging and splitting events. In our previous case studies the general evolution of the PV anomalies were captured very well. Notable and seemingly unsystematic differences, however, did occur at individual analysis times (e.g., Fig. 6 in Teubler and Riemer, 2016). Here we provide a comparison between the observed and the diagnosed amplitude tendencies in a statistical sense. Figure 2 shows scatter plots of the observed and the diagnosed amplitude evolution of all data considered in the subsequent analysis. For a perfect agreement between the observed and the diagnosed tendencies all data points would be located on the black solid line through the origin with a slope of 1. The data exhibits scatter around this line and we consider the overall agreement to be

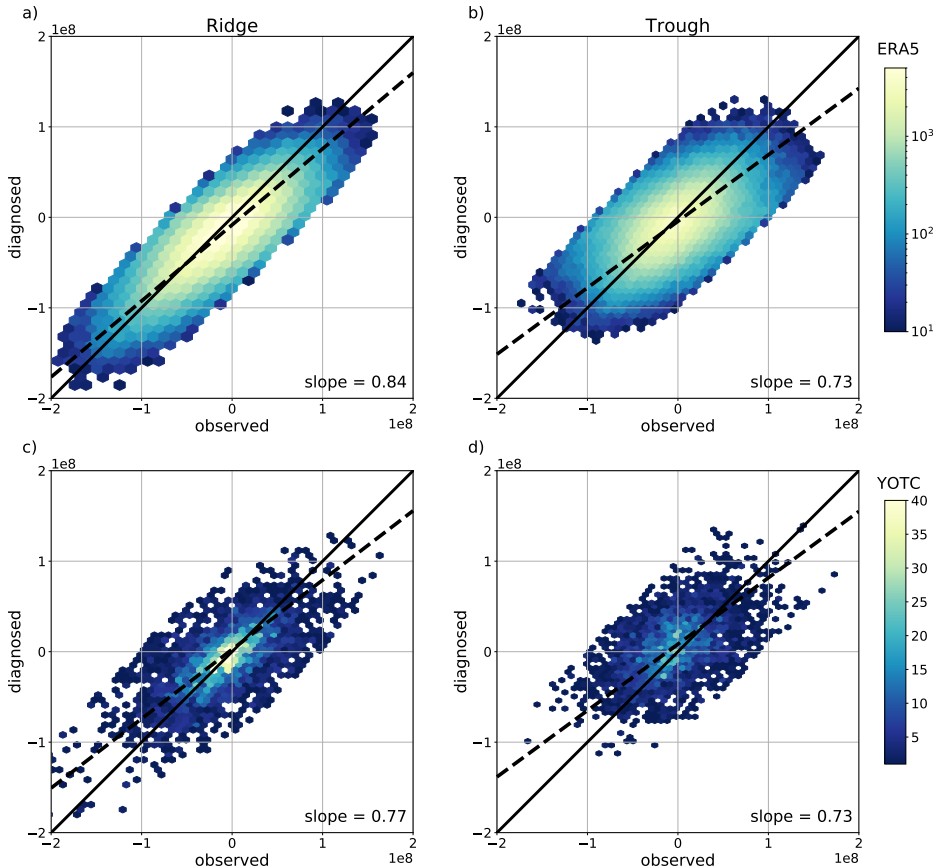

**Figure 2.** Relation between observed and diagnosed amplitude tendencies (in PVU m$^2$/s) for both datasets, ERA5 (a,b) and YOTC (c,d), and for ridges (a,c) and troughs (b,d) separately. For a perfect correspondence all scatters should be aligned along the black solid line of slope 1 and cross the origin. The 2d-fit (total least squares) is indicated by the black dashed line (slope lower right corner). Color shading describes number of values within a certain hexbin (for ERA5 on logarithmic scale). For ERA5 620316 datapoints are included, for YOTC 7692.

reasonably good, in particular for the ridges. Note that a logarithmic scale is used for the ERA5 data (Figure 2a,b) to make visible the full distribution of the data.

The actual linear fit to the data (black dashed line) exhibits an offset from the origin and a slope $< 1$. This reduced slope indicates that the diagnosed tendencies underestimate the absolute value of the observed tendency, i.e., both amplification and weakening of anomalies are underestimated. This underestimation is consistent with the fact that our diagnostic does not capture amplification by merging and weakening by splitting. Sensitivity tests have shown that increasingly more restrictive criteria for eliminating data due to differences between observed and diagnosed tendencies (by the $n$-IQR rule, see above) yield increasingly steeper (closer to 1) slopes of the linear fit in Figure 2 (not shown). Understanding that the elimination of data affects predominantly partial merging and splitting events, we have chosen our threshold $n = 1.5$ in the subsection above to be less restrictive than a standard choice of $n = 3$ to keep as many data points as possible in the ensuing analysis.

The offset from the origin is more pronounced for ERA5 than for YOTC data and is consistent with the lack of nonconservative tendencies in ERA5 data. As we will show below, nonconservative PV tendencies are on average positive leading to a systematic weakening of ridge amplitude and an amplification of trough amplitude. Consequently, the offset of the diagnosed tendencies in the ERA5 data, which comprise advective tendencies only, is negative for both ridges (Figure 2a) and troughs (Figure 2b). For the YOTC data, there is a slight offset for trough tendencies (Figure 2d) and a negligible offset for ridges (Figure 2c).

## 4 Spatial structure of PV tendencies within troughs and ridges

This section presents the spatial structure of advective PV tendencies for troughs and ridges that are part of RWPs in the northern hemisphere in the ERA5 data. Spatial composites with respect to the center of mass of the trough and ridge anomalies, respectively, are presented. We first examine the tendencies averaged over trough and ridge life cycles and discuss variations between summer and winter. These composites reveal expected characteristics of RWP propagation and baroclinic growth. In addition, we find distinct differences between troughs and ridges in a proxy for latent heat release and in the impact of the divergent flow. We argue that these differences contribute to the well-know asymmetry of troughs and ridges. Subsequently, we examine the individual tendencies in some more detail and present the spatial structure at the times when the individual tendencies exhibit their maximum and minimum values, respectively, during ridge and trough life cycles.

### 4.1 General aspects and seasonal variation

The spatial structure of the individual tendencies for the extended summer and winter seasons (May - September and November - March, respectively) are presented in Figure 3. The spatial composites comprise all troughs and ridges at all times in all RWPs in our database. As a first observation we note that the composite PV-anomaly pattern indicates an average wavenumber of 6 to 7 (wavelength of approximately $50°$) of the RWPs in both seasons.

The quasi-barotropic PV tendencies (blue contours) exhibit the pattern expected based on PV thinking for linear Rossby waves (e.g., Fig. 17 in Hoskins et al., 1985), i.e., the tendencies and the PV anomalies are in quadrature. This pattern yields a westward shift of the anomalies and thus signifies the well-know westward intrinsic phase velocity of Rossby waves. In addition, in a spatially-integrated sense, the quasi-barotropic tendencies in the region of the downstream anomalies amplify these anomalies. In contrast, upstream anomalies weaken. This pattern of quasi-barotropic tendencies implies a downstream propagation of the wave packet and thus signifies the eastward intrinsic group velocity of RWPs. Both tendency patterns do not exhibit notable variations between ridges and troughs and between summer and winter.

The baroclinic PV tendencies (yellow contours) clearly demonstrate that RWPs amplify on average due to baroclinic growth. Positive baroclinic tendencies spatially correlate on average with positive PV anomalies and negative tendencies with negative anomalies, thus amplifying the existing anomalies. Consistently, the low-level temperature pattern exhibits an according phase shift. Baroclinic growth in summer is much weaker than in winter (about 50%) associated with a 25-30% weaker low-level temperature gradient in summer (not shown).

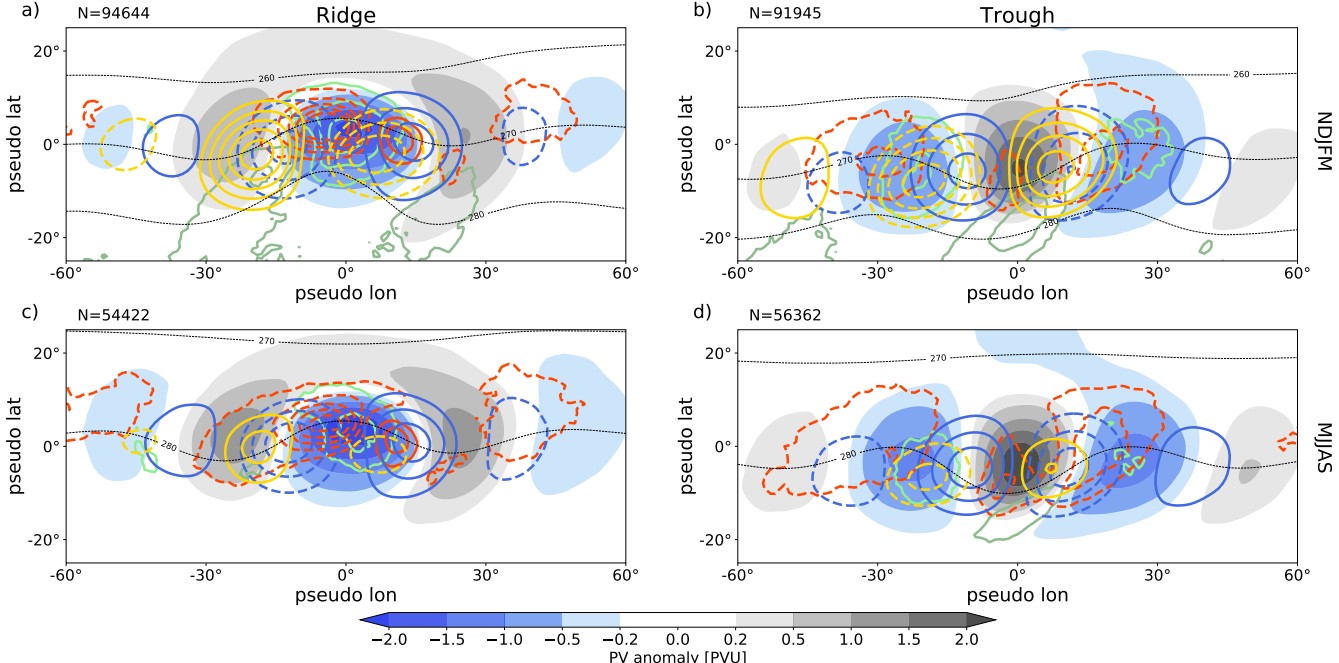

**Figure 3.** Spatial composites of PV anomalies (shading) with individual PV′ tendencies (contours) for both NDJFM (a,b) and MJJAS (c,d) and for ridges (a,c) and troughs (b,d) separately. The different contours show in light green convergence of IVT (1000-500 hPa) if relative humidity $\geq 80\%$ as proxy for latent heat release in the lower to mid-troposphere ($\pm$(0.02, 0.04, 0.06, 0.08, 0.1) kg m$^{-2}$ day$^{-1}$, divergence of IVT in dark green), in blue PV tendencies due to quasi-barotropic propagation ($\pm$(1, 2, 3, 4, 5) PVU/day), in yellow baroclinic interaction ($\pm$(0.1,0.15,0.2,0.25,0.3) PVU/day), in red PV tendencies due to divergent flow ($\pm$(0.2, 0.4, 0.6, 0.8, 1) PVU/day) and in thin black mean temperature between 850 and 800 hPa ((260, 270, 280) K). Dashed contours of PV tendencies refer to negative values, respectively. Note, the mean temperature indicates the location of warm (northward extension of contour) and cold (southward extension of contour) anomalies at lower levels. The isentropic level for PV anomalies and PV tendencies follows the seasonal cycle.

The PV tendencies due to the divergent flow (red contours) exhibit distinct differences between troughs and ridges. These tendencies are predominantly negative and maximize within the ridges. Thereby, on average, the divergent flow amplifies ridges (Figure 3a,c) and, to lesser extent, weakens troughs (Figure 3b,d). Within the ridges, the divergent tendency exhibits a dipole with negative tendencies ahead of the upstream trough and positive tendencies in the rear of the downstream trough (Figure 3a,c). It is plausible that this dipole is related to the dry dynamics of a baroclinically growing wave, in which ascent and upper-tropospheric divergence and their impact on the PV field occurs ahead of troughs and descent, upper-tropospheric convergence, and the respective impact in the rear of troughs (e.g., Fig. 8.10 in Holton, 2004). The negative tendencies of the dipole clearly dominate the positive tendencies, consistent with the invigoration of ascent and thus upper-tropospheric divergence by latent heat release below, i.e., in the lower to mid-troposphere. To examine the role of lower to mid-tropospheric latent heat release, we consider as a proxy the convergence of integrated water vapor transport (IVT) if relative humidity is

larger than 80%, vertically integrated from $1000 - 500\,\mathrm{hPa}$[5], similar to previous studies (e.g., Berman and Torn, 2019). This proxy (light green contours) demonstrates systematic release of latent heat within the warm anomaly underneath the ridge (Figure 3a,c). Arguably, this latent heat release is associated with the warm conveyor belts of extratropical cyclones. In winter, latent heat release is approximately 50% stronger than in summer, consistent with stronger baroclinic development in winter. The divergent tendencies, however, show substantially less differences between summer and winter. In contrast, there is on average no indication of latent heat release within troughs (Figure 3b,d). Instead, divergence of IVT is found equatorward of the troughs within relatively cold air. This signal is arguably associated with evaporation in descending air masses and surface moisture fluxes in the cold sector of cyclones.

## 4.2 Extrema of individual tendencies during trough and ridge life cycles

This subsection examines the individual tendencies in some more detail. Specifically, we consider the composite spatial structure at the time in the trough and ridge life cycles at which the individual advective tendencies (the first three terms in Equation 6) exhibit their respective maximum and minimum spatially- integrated value. Recall that the amplitude metric in this study is defined as the integral of $PV'$ over the area of the anomaly (subsection 2.3). The composites include cases only if the maximum value during the life cycle is positive and the minimum value is negative, which is the case for 70-90% of the troughs and the ridges. Note that we select the extrema of the individual tendencies in this section again year-round.

Maximum amplification by the quasi-barotropic tendency is very similar for troughs and ridges and occurs when the upstream PV anomaly is substantially stronger than the downstream anomaly (Figure 4a,b). This configuration leads to larger amplifying tendencies on the upstream side of the respective ridge or trough than weakening tendencies on its downstream side, and thus overall amplification in the spatially-integrated sense. The opposite is true for the strongest weakening of anomalies by the quasi-barotropic tendency (Figure 4c,d). This configuration of the PV anomalies signifies that maximum amplification and strongest weakening occur towards the leading and the trailing edge of the RWP, respectively. This quantitative result derived from all RWPs during the ERA5 period from 1979-2017 confirms expectations based on linear wave packet dynamics. Note, however, that our diagnostic does not capture decay due to wave breaking, i.e., highly nonlinear evolution. In general, the amplitude change due to the quasi-barotropic tendency depends on the degree of asymmetry between the anomalies up- and downstream of the anomaly of interest. The largest individual amplifications in our data (top 5%, not shown) occur when there is, e.g., a pronounced trough upstream of a developing ridge but no downstream trough. Instead, e.g., a further weak ridge occurs in the farther downstream region that is apparently unrelated to the RWP under consideration (analogous for strongly amplifying troughs). Based on this observation we conclude that the most extreme amplification of troughs and ridges in RWPs due to quasi-barotropic dynamics occurs when the RWP interacts with preexisting like-signed PV anomalies in the downstream region.

The characteristics of maximum and minimum baroclinic tendencies are also very similar for troughs and ridges (Figure 5). Note that we here consider the absolute growth of anomalies rather than growth rates, which are often considered in more theoretical studies of baroclinic instability. The largest baroclinic amplification here occurs when the respective anomaly is already

---

[5]Convergence of IVT is proportional to latent heat release for saturated conditions and in the absence of ice processes.

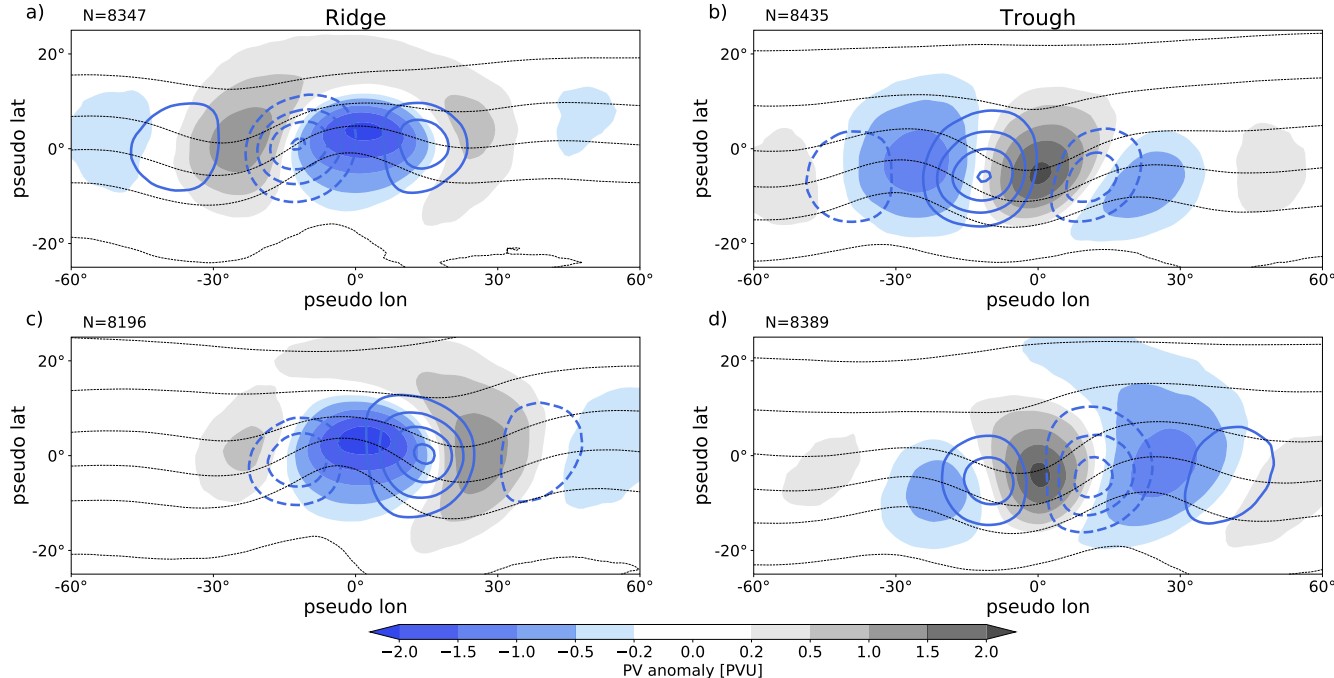

**Figure 4.** Spatial composites of PV anomalies (shading) and PV′ tendencies for (a,c) ridges and (b,d) troughs at times when the quasi-barotropic PV tendencies yield maximum amplification (a,b) and maximum weakening (c,d) of the respective amplitude. The different contours show in blue PV tendencies due to quasi-barotropic propagation ($\pm$(1, 2, 3, 4, 5) PVU/day, negative dashed) and in thin black mean temperature between 850 and 800 hPa (every 5 K). Note, the mean temperature indicates the location of warm (northward extension of contour) and cold (southward extension of contour) anomalies at lower levels. The isentropic level for PV anomalies and PV tendencies follows the seasonal cycle.

of large amplitude and accompanied by a downstream anomaly that is stronger than the upstream anomaly (Figure 5a,b). This pattern signifies that largest baroclinic amplification occurs on average during the mature stage of individual troughs and ridges and rather towards the trailing edge of the RWP. Our analysis of a large number of cases thus confirms this aspect of a concep-

380 tual model presented in a recent review of RWP dynamics (Wirth et al., 2018, their Fig. 9). The low-level temperature wave exhibits a favorable phase shift and the baroclinic tendencies are almost in phase with the upper-tropospheric PV anomalies. A distinct difference between troughs and ridges is found in the proxy for latent heat release, which indicates that strong baroclinic amplification of ridges is associated with strong latent heat release. An analogous signal for troughs is not discernable. Weakening by the baroclinic term occurs on average towards the leading edge of the RWP, with more pronounced upstream

than downstream anomalies (Figure 5c,d). Prominent baroclinic tendencies occur upstream of the respective anomaly only, consistent with the lack of a prominent downstream temperature perturbation at the leading edge of the RWP. These tendencies favor baroclinic amplification of the upstream anomaly but weaken the anomaly on that the composites are centered. The lack of compensating baroclinic tendencies from temperature anomalies in the downstream region make plausible why weakening

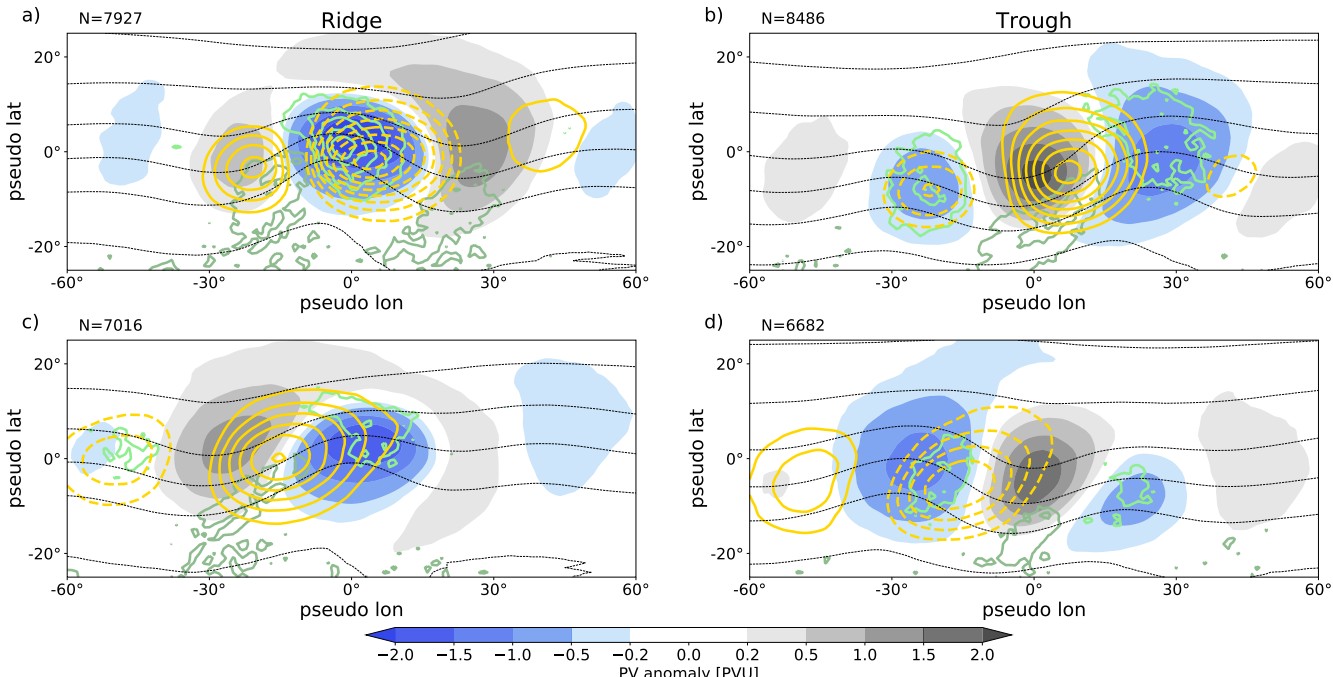

**Figure 5.** Spatial composites of PV anomalies (shading) and PV′ tendencies for (a,c) ridges and (c,d) troughs at times the baroclinic PV tendencies yield maximum amplification (a,b) and maximum weakening (c,d) of the respective amplitude. The different contours show in yellow PV tendencies due to baroclinic interaction (±(0.1, 0.15, 0.2, 0.25, 0.3, 0.35, 0.4, 0.45, 0.5) PVU/day, negative dashed), in light green convergence of IVT as proxy for latent heat release (±(0.02,0.04,0.06,0.08,0.1) kg m$^{-2}$day−1, divergence of IVT in dark green) and in thin black mean temperature between 850 and 800 hPa (every 5 K). Note, the mean temperature indicates the location of warm (northward extension of contour) and cold (southward extension of contour) anomalies at lower levels. The isentropic level for PV anomalies and PV tendencies follows the seasonal cycle.

by baroclinic interaction preferentially occurs near the leading edge of RWPs. Note that there is substantially less latent heat release within ridges that weaken by baroclinic interaction compared to those that amplify (cf. Figure 5a,c).

The impact of the divergent flow and associated moist processes exhibit the most distinct differences between troughs and ridges (Figure 6). The absolute value of the divergent tendency is by far the largest for ridge amplification. During maximum ridge amplification, strong upper-level divergence occurs mostly within the ridge anomaly, clearly associated with large values of the proxy for latent heat release below (Figure 6a). Latent heat release occurs on average just downstream of an upstream surface cyclone, which indicates that latent heat release occurs within the warm conveyor belt of that cyclone. The amplitude of the upper-level PV anomalies and the existence of well-developed surface systems demonstrate that the strongest divergent ridge amplification occurs on average during the mature stage of RWPs. The difference in amplitude between the up- and downstream troughs is less pronounced than during maximum baroclinic ridge amplification (cf. Figure 6a and Figure 5a), which signifies that maximum divergent amplification occurs on average closer to the center of RWPs and thus in advance of the

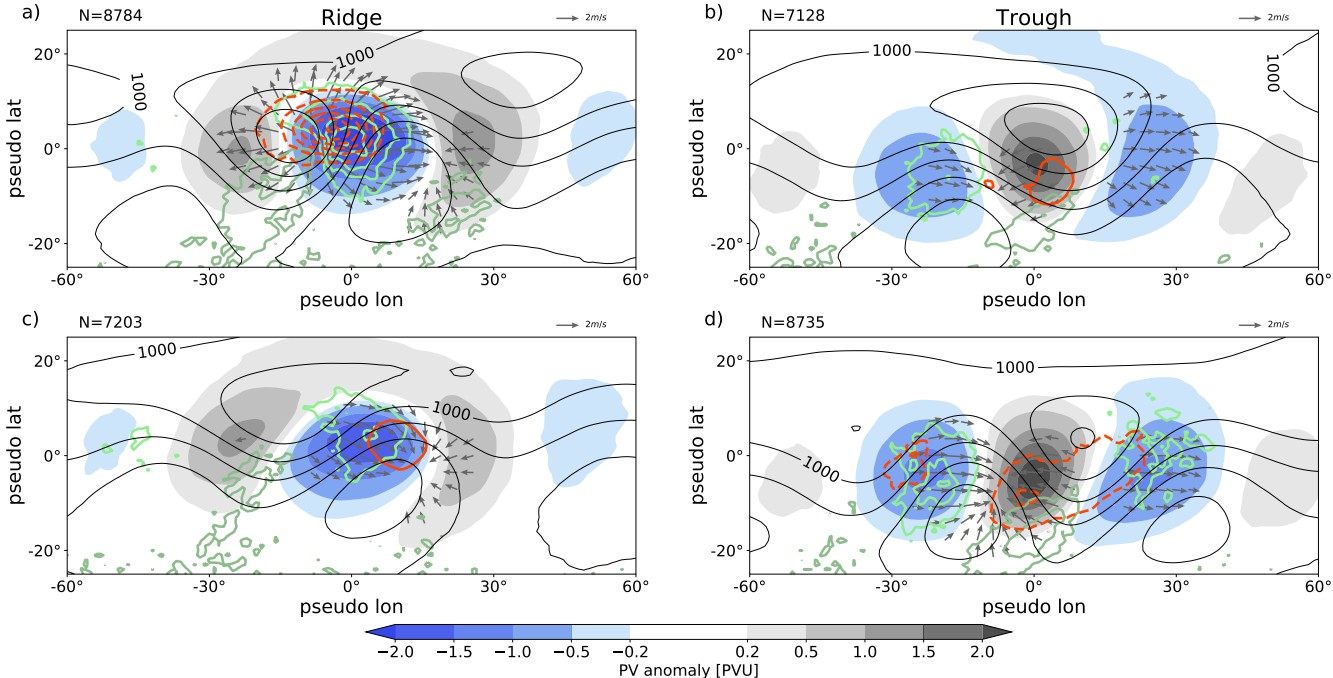

**Figure 6.** Spatial composites of PV anomalies (shading) and PV′ tendencies for (a,c) ridges and (b,d) troughs at times when the divergent PV tendencies yield maxmimum amplification (a,b) and maximum weakening (c,d) of the respective amplitude. The different contours show in red PV tendencies due to divergent flow ($\pm$(0.5, 1, 1.5, 2, 2.5, 3) PVU/day, negative dashed), in light green convergence of IVT as proxy for latent heat release ($\pm$(0.02,0.04,0.06,0.08,0.1) kg m$^{-2}$day$-1$, divergence of IVT in dark green) and in thin black geopotential at 1000 hPa (every 200 m$^2$/s$^2$, 1000 m$^2$/s$^2$ labeled). The arrows refer to the divergent wind, only shown at grip points with values greater than $1\,m/s$ (reference vector in upper-right corner). The isentropic level for PV anomalies and PV tendencies follows the seasonal cycle.

maximum of baroclinic amplification. In contrast, maximum weakening of ridges by the divergent flow tends to occur towards the leading edge of RWPs (Figure 6c). The upper-tropospheric divergent flow in this case is dominated by convergence on the downstream side of the ridge and is associated with substantially less latent heat release than during maximum amplification (cf. Figure 6a). For troughs, both, maximum divergent amplification and weakening occur, on average, near the center of the RWP (Figure 6b,d). Differences between amplification and weakening are most prominently found in the characteristics of the low-level pressure systems and of the proxy for latent heat release: i) The low-level geopotential exhibits a higher amplitude pattern, ii) the location of the center of the downstream cyclone is below the trough, and iii) moist processes are more pronounced for maximum weakening (Figure 6d) than for amplification (Figure 6b). In addition, the upper-tropospheric divergent flow during maximum weakening exhibits stronger convergence on the upstream side of the trough, consistent with larger latent heat release underneath the upstream ridge.

In summary, our quantitative results based on a large number of RWPs confirm the existing conceptual model of RWP dynamics (Fig. 9 in Wirth et al., 2018), save highly nonlinear evolution (wave breaking), which is not captured by our diagnostic

framework. On average, intrinsic group and phase propagation are consistent with linear Rossby-wave theory. Baroclinic and divergent amplification occur preferentially near the center and towards the trailing edge of RWPs. Besides this refinement of the specific timing of maximum baroclinic and divergent amplification, we here provide for the first time a comprehensive analysis of the role of the divergent flow for trough and ridge amplitude. Distinct differences between troughs and ridges are demonstrated, which will be elaborated on below. The question to what extent divergent ridge amplification is associated with latent heat release will further be addressed below also (subsection 5.3).

## 5 Temporal evolution and sequence of governing mechanisms

After discussing their spatial distribution, this section investigates the temporal evolution of and relationship between the individual PV tendencies. For a succinct depiction of temporal characteristics, the individual tendencies are spatially integrated over the area of the respective PV anomaly. To simplify the presentation, the sign convention will be such that positive tendencies indicate amplification and negative tendencies indicate weakening for both, troughs and ridges. Since the composites made with ERA5 and YOTC look very similar (not shown), we show here the advective tendencies from the ERA5 data and add the nonconservative PV tendencies calculated from the YOTC data. A strong interconnection between divergent ridge amplification and both, baroclinic growth and latent heat release will be demonstrated, hinted at already above. The last subsection will make first strides towards disentangling contributions of moist and dry (balanced) dynamics to this divergent amplification.

### 5.1 Individual mechanisms in relation to maximum amplitude

We first focus on amplitude evolution and thus consider composites with respect to the maximum amplitude of troughs and ridges, respectively (Figure 7). Some of the general features of the evolution are similar for both, troughs and ridges, and for both, winter and summer. By design, the observed amplitude change is positive before and negative after the anomalies' maximum amplitude. Interestingly, the transition from strongly positive to strongly negative tendencies occurs rapidly, on a timescale of 12 h. Partly, this feature is an artefact of the compositing technique: With increasing distance from the composite time the composite average converges towards a climatological value. The largest absolute values of the composite mean, and thus the sharpest gradients, can therefore be expected close to the composite time. Partly, however, we consider the rapid transition to be physically meaningful and indicative of a rapid onset of the decay of anomalies after they have reached their maximum amplitude. The decay is dominated by the quasi-barotropic term changing from positive to negative values. The rapid transition is well represented by the diagnosed PV tendencies in the ridge composite, but less so in the trough composite (cf. Figure 7a,c and Figure 7b,d). Our interpretation of this discrepancy is that (partial) wave breaking and associated splitting of the PV anomaly after reaching maximum amplitude, which is not captured well by our diagnostic (see discussion in Sect. 3.3), occurs more prominently for troughs than for ridges. A further general similarity between troughs and ridges and summer and winter is that the baroclinic term exhibits on average the smallest absolute value of the advective tendencies.

In winter, amplitude evolution follows the paradigm of downstream baroclinic development: on average, troughs and ridges first grow by downstream development, i.e., the quasi-barotropic term, followed by a maximum of baroclinic growth approx-

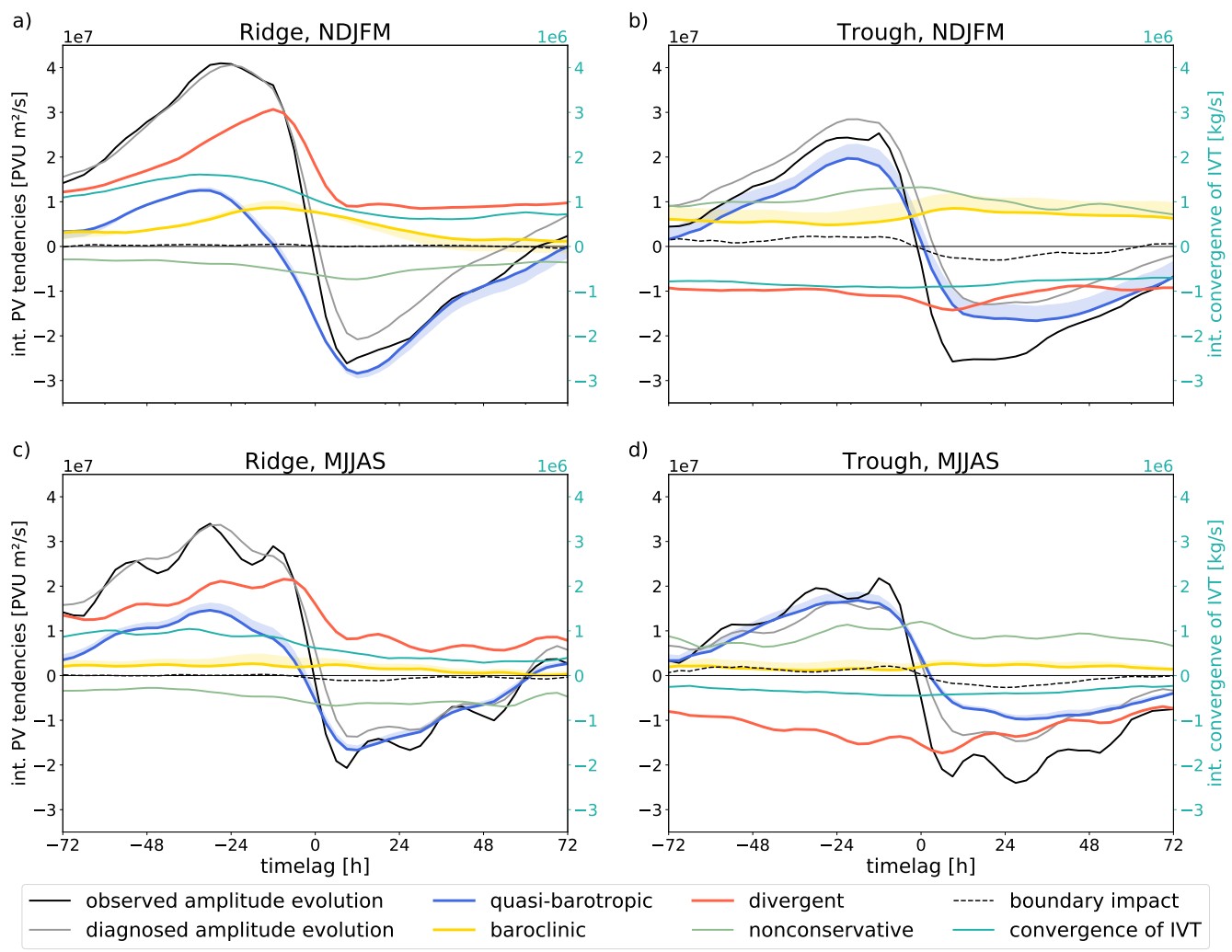

**Figure 7.** Time series of the individual contributions (left y-axis) to the evolution of the (a,c) ridge- and (b,d) trough-composite for NDJFM (a,b) and MJJAS (c,d), respectively. Convergence of IVT (right y-axis) is integrated over the same area as PV tendencies. The x-axis depicts the time lag in hours relative to the time of maximum amplitude. The sign of the tendencies is defined such that positive (negative) values always indicate amplification (weakening) of the composite-anomaly, regardless if trough or ridge. Only shown between lag=-72 h and lag=+72 h around the time of maximum amplitude. The shading around the quasi-barotopic and baroclinic contribution is given by $\mathbf{v_{res}}$ from Equation 4 and indicates the uncertainty of piecewise PV inversion. A running mean of 3 time steps is applied on processes to smooth curves. Nonconservative tendencies are included from YOTC-data (available every 6 h, compared to every 3 h from ERA5-data).

imately one day later (Figure 7a,b). This signal is not clearly evident in summer, when the baroclinic term is substantially

smaller than in winter (cf. (Figure 7c,d and Figure 7a,b), consistent with the seasonal differences in the spatial composites discussed above (Figure 3). While reduced baroclinic growth during summer is not an unexpected results, the new aspect here is that the baroclinic term in summer exhibits on average much less relation to the amplitude evolution than in winter. The occurrence and the characteristics of downstream baroclinic development in the composites will be investigated in some more detail in Sect. 5.2.

A difference in the evolution of troughs and ridges is that the maximum amplitude of ridges occurs on average when the quasi-barotropic term has already turned negative, whereas the maximum amplitude of troughs occurs while this term is still positive. This difference arises because the divergent term is positive for ridges but negative for troughs (cf. Figure 7a,b and Figure 7c,d). This consistent amplification and weakening by the divergent term, respectively, is the most striking difference between ridges and troughs, in both summer and winter, consistent with the results from the spatial composites discussed in

section 4.

The impact of the divergent flow is dominated by the term $PV'(\nabla \cdot \mathbf{v}_{div})$ in Equation 6 (not shown), i.e., ridge amplification implies an increase of the spatial scale of the ridge anomaly and the weakening of troughs implies a decrease of the spatial scale of the trough anomaly. Differences in the spatial scale of troughs and ridges are a well known feature and can be explained to lowest order by (dry) semi-geostrophic theory (Hoskins, 1975; Wolf and Wirth, 2015). Semi-geostrophic theory extends quasi-

460 geostropic theory by including the ageostrophic wind in the advection of geostrophic momentum, from which the asymmetry between troughs and ridges eventually arises. The divergent flow contributes to the ageostrophic wind. While dry theory explains the ridge-trough asymmetry to lowest order, our results show that other processes that lead to upper-level divergence, most notably latent heat release, contribute further to the observed asymmetry.

The maximum amplitude of ridges is on average associated with a maximum of the divergent term (Figure 7a,c), indicating

that the maximum amplitude of ridges is strongly related to ridge building by upper-level divergent outflow. This mechanism for ridge amplification has been given much attention in the literature (see references in the introduction). In contrast, the impact of the divergent flow on troughs has, to our knowledge, previously been examined only for few individual cases (e.g., Pantillon et al., 2013; Teubler and Riemer, 2016) and in an idealized scenario (Riemer and Jones, 2014). The strong weakening of troughs after their maximum amplitude is associated with a minimum in the divergent term (Figure 7c,d), indicating that the

divergent flow has a particularly detrimental impact on trough amplitude during this part of a trough's life cycle. Our results thus provide evidence that, on average, upper-level divergent flow contributes systematically to the amplitude evolution of troughs also.

The absolute value of the total nonconservative tendencies is similar to that of the individual advective tendencies, in both summer and winter (Figure 7). These tendencies are clearly dominated by radiation (Figure 8 and Figure 9). Other nonconser-

475 vative tendencies, including those due to latent heat release, are an order of magnitude smaller than the advective tendencies. Radiative tendencies weaken ridges and strengthen troughs as was noted before, e.g., by Zierl and Wirth (1997) for idealised anyticyclonic upper-level PV anomalies and by Chagnon et al. (2013) who showed positive PV tendencies within the stratosphere and especially in troughs. The absolute value of the radiative tendency correlates with trough and ridge amplitude

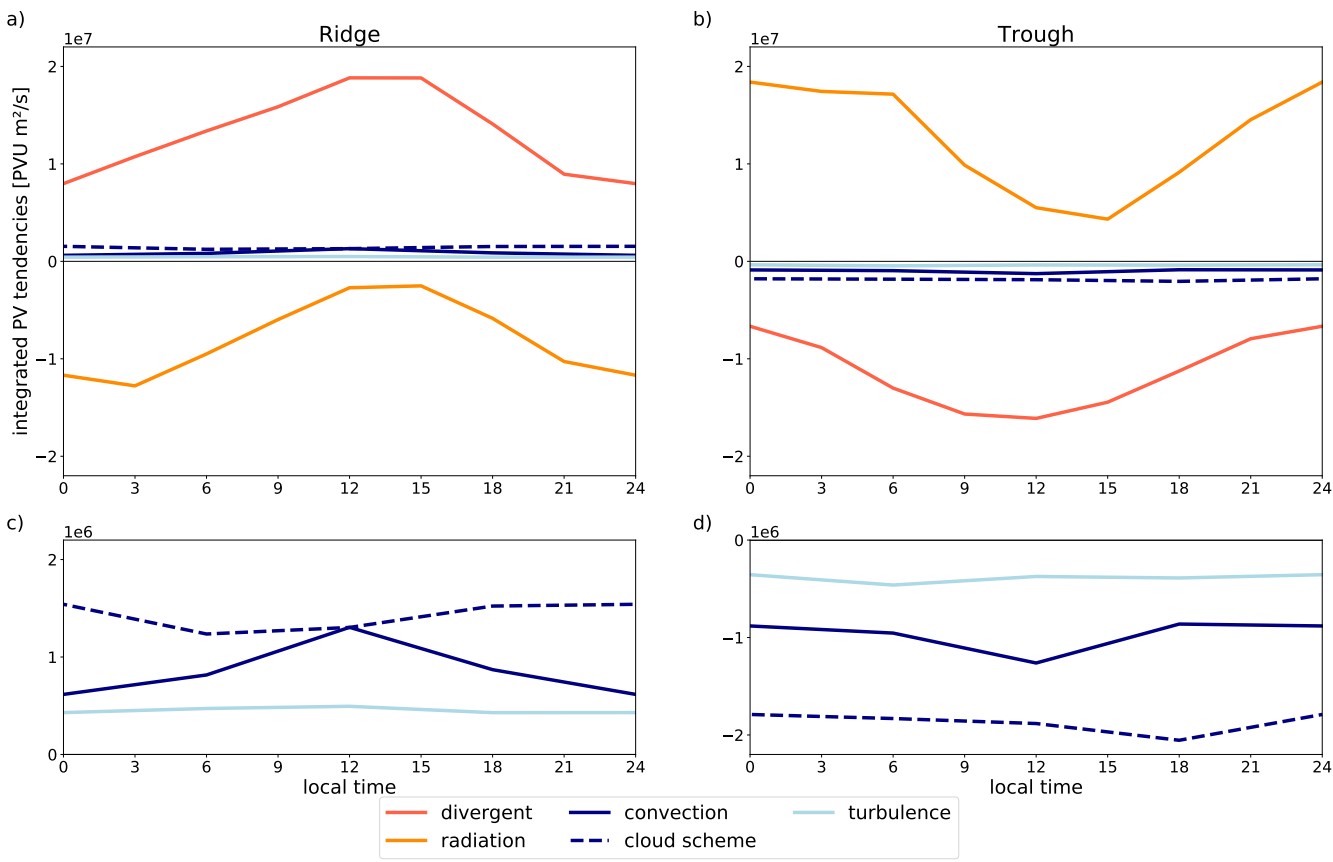

**Figure 8.** Diurnal cycle of PV′ tendencies in MJJAS for both ridges (a,c) and troughs (b,d). (c,d) zoom into the nonconservative PV tendencies for better visualization of its diurnal cycle. Local time sorted relative to the center of mass of ridges and troughs and downscaled to a 3 hourly (ERA5) and 6 hourly (YOTC) resolution.

(indicated in Figure 7 but not shown explicitly). This correlation arises because the radiative tendencies are dominated by the

term $\dot{\theta} \partial PV / \partial \theta$, i.e., cross-isentropic transport of PV by longwave radiative cooling (not shown). Because $PV = \overline{PV} + PV'$ and, by our definition, $\overline{PV}$ does not vary over the life time of an RWP, this correlation with amplitude $PV'$ can be expected.

Intriguingly, in the summer composites (Figure 7c,d), the observed amplitude evolution and, to lesser extent, the divergent term exhibit a weak diurnal cycle. Figure 8 depicts the divergent tendency and the individual nonconservative tendencies during summer as a function of local time. The divergent, longwave radiative, and convective tendencies each exhibit a clear

diurnal cycle with maxima for troughs and minima for ridges around noon and in the early afternoon. We thus argue that the diurnal cycle in the observed amplitude tendencies is a combination of the direct radiative (solar) cycle and the diurnal cycle of convection, in which convection impacts i) the divergent flow by latent heat release and ii) radiation by cloud formation and changes in upper-tropospheric humidity. Gristey et al. (2018) have shown that the impact of convection on radiation modifies the diurnal cycle of longwave radiation, which is predominantly governed by the diurnal cycle of land surface heating, and

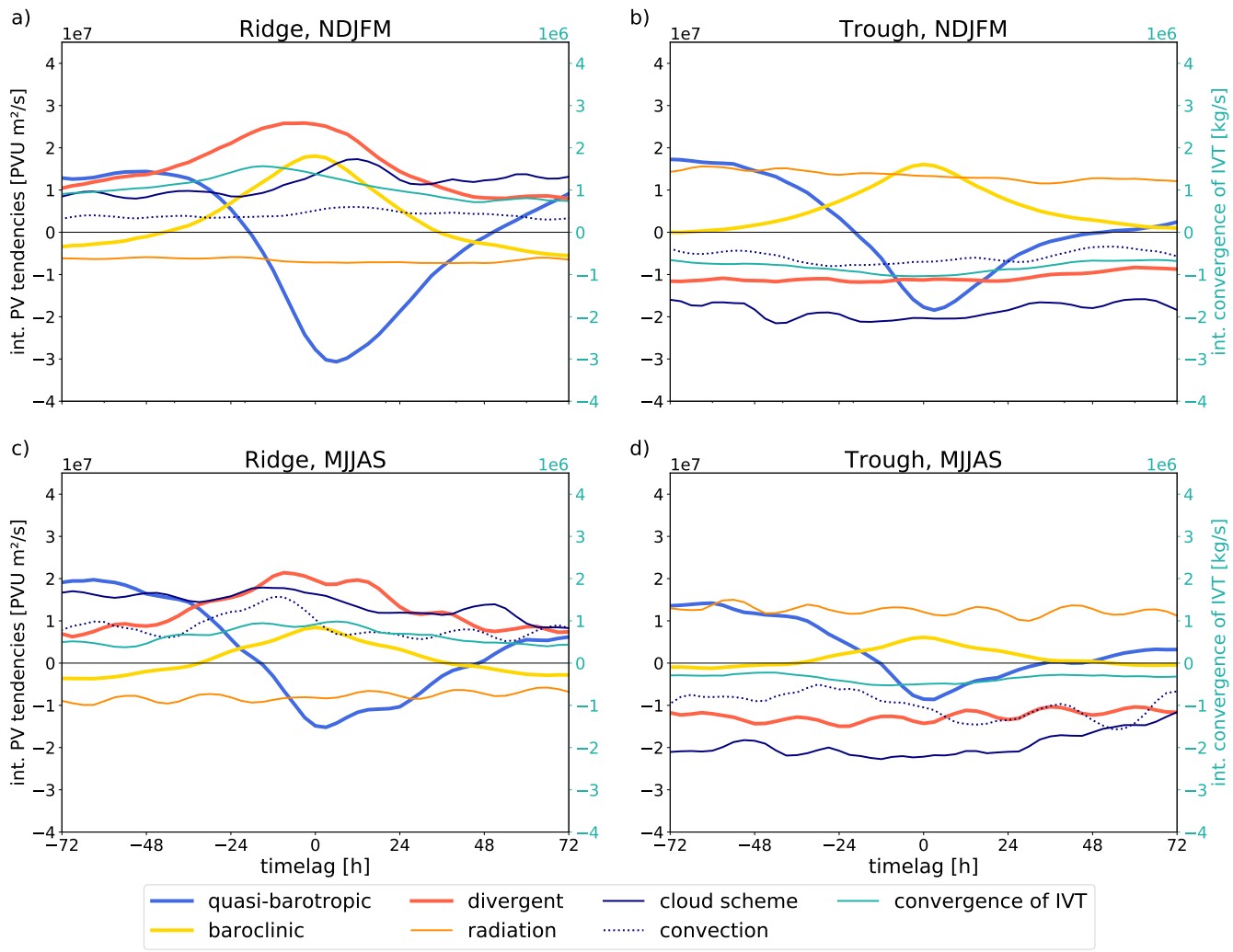

**Figure 9.** Same as Figure 7, but x-axis relative to the maximum of baroclinic growth. Contributions due to convection and cloud scheme multiplied by factor 10 for better visualization.

the associated emission temperature, by insolation. The diurnal cycle in the PV tendencies imprints on the mean-amplitude evolution because maximum amplitude, on which the composites are centered, is more likely to be reached when the diurnal cycle is in a favorite phase. A more detailed discussion of the processes that govern the observed diurnal cycle is beyond the scope of the current study. However, we will consider the horizontal and vertical structure of the longwave radiative PV tendencies below when discussing Figure 10.

## 5.2 Downstream moist-baroclinic development

The composites discussed above (Figure 7) clearly demonstrate the validity of the paradigm of downstream baroclinic development for the evolution of RWPs in winter. It is not clear from those composites, however, if this paradigm provides also a reasonable description for RWP evolution in summer. This subsection addresses this question by focusing more directly on how baroclinic growth is embedded in the sequence of governing mechanisms, i.e., we here consider composites not with respect to maximum amplitude, as above, but with respect to the governing processes themselves. Specifically, we consider the baroclinic life cycle of troughs and ridges by centering the composites on the time of maximum baroclinic growth (Figure 9). These composites confirm that the mean temporal evolution in winter follows the paradigm of downstream baroclinic development (Figure 9a,b): 1-2 days before maximum baroclinic growth both, troughs and ridges, amplify by the quasi-barotropic tendency, i.e., downstream propagation. This tendency turns distinctly negative during prominent baroclinic growth and remains negative for at least 1 day after maximum baroclinic growth. The baroclinic life-cycle composites now reveal that the same sequence of processes occur also in summer (Figure 9c,d). The paradigm of downstream baroclinic development thus provides a valid description of the mean evolution of troughs and ridges in summer. The magnitude of baroclinic growth, however, is only half of that in winter, consistent with Figure 7.

Our proxy for latent heat release systematically varies during the baroclinic life cycle. Ridges exhibit maxima during prominent baroclinic growth with values that are about 50% (winter, Figure 9a) and 100% (summer, Figure 9c) higher than when the baroclinic tendency is negative. Ridges in winter exhibit a relatively sharp maximum in the proxy for latent heat release that occurs 12-18 h before maximum baroclinic growth (Figure 9a). Troughs exhibit minima with values that are about 30% (winter, Figure 9b) and 60% (summer, Figure 9d) lower than when the baroclinic tendency is relatively small. These systematic relationships demonstrates the coupling of moist and baroclinic processes in midlatitude RWPs. Most striking, however, is the strong correlation of the baroclinic and the divergent tendency for ridges: A clear maximum in the divergent term occurs about 12 h before maximum baroclinic growth (Figure 9a,c). In contrast, there is no such systematic relation for troughs (Figure 9b,d)[6]. While the divergent term is related to trough amplitude (Figure 7), the detrimental impact of the divergent term does not vary systematically during the trough's baroclinic life cycle. Ridge building by the divergent flow, in contrast, is evidently strongly coupled to moist-baroclinic development and, consistent with Figure 7, makes a first order contribution to amplitude evolution during the baroclinic life cycle of ridges, both in winter and in summer.

The relation between nonconservative terms and the baroclinic life cycle is less clear. In winter, maxima in ridge amplification by the convection and the cloud scheme occur 12-18 h after maximum baroclinic growth (Figure 9a) and are respectively 10% and 60% stronger than when baroclinic tendencies are negative. Interestingly, these maxima of PV tendencies occur distinctly later than the maximum of our proxy for latent heat release. In summer, ridge building due to the convection scheme exhibits a distinct maximum about 12 h before maximum baroclinic growth (Figure 9c) and is 60% stronger than when baroclinic tendencies are negative. For troughs (Figure 9b,d), it is less clear how variations of tendencies due to the convection and

---

[6]Centering the composites on the time of maximum absolute value of the divergent term yields consistent results.

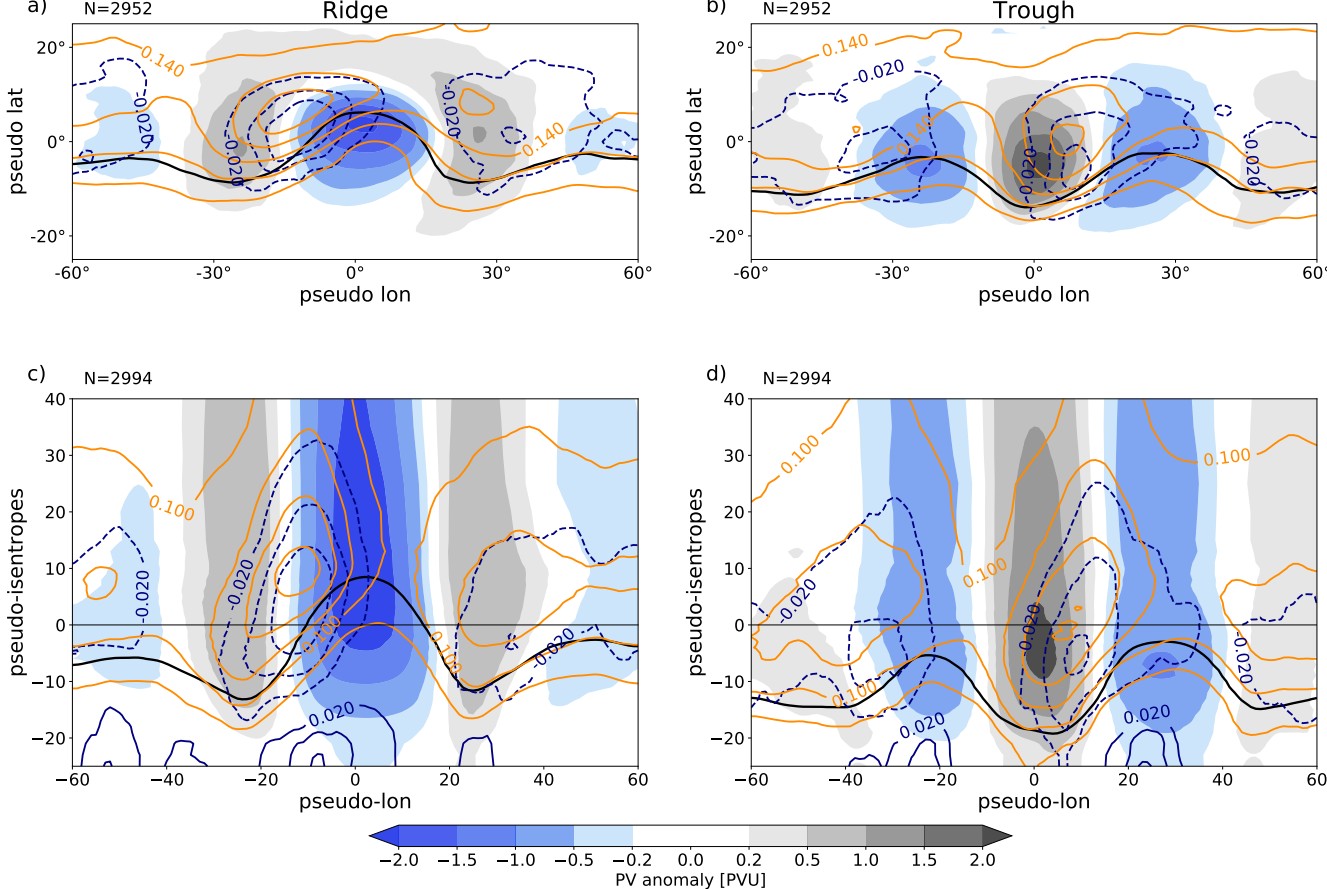

**Figure 10.** Spatial composites of PV anomalies (shading) and PV' tendencies for (a,c) ridges and (b,d) troughs during YOTC-period averaged from 72 h before to 72 h after the maxmimum of baroclinic growth. The different contours show in blue PV tendencies due to the cloud scheme ($\pm(0.02, 0.04, 0.06)$ PVU/day, negative dashed), in orange PV tendencies due to longwave radiation ($\pm(0.06, 0.1, 0.14, 0.18, 0.22)$ PVU/day) and in black the 2-PVU contour depicting the tropopause. The isentropic level in (a,b) follows the seasonal cycle defined in subsection 2.2 and relates to the pseudo-isentropic level 0 in (c,d). Note, only YOTC-data is shown. A box-smoothing of 3 points is applied for visual clarity.

cloud scheme relate to the baroclinic life cycle. As noted above, these nonconservative tendencies are an order of magnitude smaller than the advective tendencies and thus make only a minor contribution overall.

The longwave radiative tendency is largely constant during the baroclinic life cycle of both, ridges and troughs. Because this tendency makes a contribution to the amplitude evolution comparable to the advective tendencies, and due to the large recent interest in cloud-radiative feedback on extratropical storm tracks and on cyclone evolution (e.g., Schäfer and Voigt, 2018; Grise et al., 2019; Papavasileiou et al., 2020), we investigate the radiative tendency in some more detail. Figure 10 depicts the spatial pattern of this tendency (orange contours). To lowest order, the pattern can be considered to comprise two components. The first component is characterised by isolines that are parallel to the undulated tropopause, with values increasing from the

troposphere to the stratosphere. This increase is arguably associated with the strong moisture gradient across the tropopause. Superimposed on this "background" component are local maxima ahead of troughs with values that are approximately 50% larger than the "background" values. The location of these maxima is consistent with a schematic of the typical radiative impact on the synoptic-scale wave pattern by Chagnon et al. (2013, their Fig. 10), hypothesized from the results of a detailed case study. The spatial pattern of the radiative tendencies thus reveals important variations, in contrast to the temporal evolution

of the spatially-integrated values relative to the maximum of baroclinic growth (Figure 9). In particular, the maximum ahead of troughs can be associated with a maximum in the occurrence of clouds: Large-scale ascent and associated cloud formation is usually expected ahead of a trough. Using the PV tendencies from the cloud scheme as a rather rough, but from our data easily available proxy for cloud occurrence confirms the clear relation of clouds and the extrema in radiative PV tendencies (Figure 10). Importantly, the patterns and their amplitudes depicted in Figure 10 do not vary appreciably over the average

baroclinic life cycle, as here defined from minus to plus three days around the maximum of baroclinic growth: At any stage, baroclinic development occurs within a wavy upper-level pattern with an extremum of our proxy for cloud occurrence ahead of the trough (not shown). This continued existence of an upper-level wave pattern is consistent with our selection of individual cases as being part of RWPs.

Our interpretation of the results for the radiative tendencies is that the majority of the modification of trough and ridge

amplitude is associated with "background" radiation, i.e., with radiative tendencies that are associated with the climatological feature of a strong moisture gradient across the tropopause. It seems plausible that this impact exhibits little coupling with the underlying dynamics (as found in Figure 9). A further, notable part of the radiative tendencies is apparently associated with cloud-radiative effects. There is thus the potential that cloud-radiative effects impact baroclinic development by the direct diabatic modification of upper-level PV anomalies. From the results of this study alone, however, it is not straight-forward to

compare our findings to the existing literature on cloud-radiative feedback on storm tracks. Most importantly, cloud-radiative effects on extratropical storm tracks are more complex than by direct upper-level PV modification (Grise et al., 2019). The lack of variation with the baroclinic life cycle found herein hinges on the continued existence of a wavy upper-level pattern. Examinations of different scenarios, e.g., idealized life cycles that start from a straight jet may yield a different result. In addition, the lack of variation over the composite life cycle does not exclude the potential for important differences between

individual cases, which warrants future investigations into the case-to-case variability of the cloud radiative component of direct diabatic PV modification of troughs and ridges.

## 5.3 Divergent ridge amplification and moist-baroclinic development

This section addresses the question to what extent divergent ridge amplification can be attributed to the dry (balanced) dynamics of the baroclinically-growing wave and to latent heat release below. An answer to this question is of importance because it

provides increased understanding of the sensitivities of the extent to which moist processes impact the large-scale flow by associated upper-tropospheric divergence. For the sake of brevity, we will abbreviate the divergent tendency, the baroclinic tendency, and the proxy for latent heat release in this subsection by DIV, BC, and LHRproxy, respectively. The basic idea is to

use our data to attribute DIV to LHRproxy and to BC, respectively, with BC here serving as a simple proxy for the state of the baroclinic development and thus as a proxy for the characteristics of the dry (balanced) dynamics.

First we note that ridge amplification by DIV is related to both, increased BC and increased LHRproxy (Figure 9a,c). Spatial composites at the time of maximum ridge amplification by DIV clearly show increased LHRproxy (Figure 6a) and it is clear from the composites at the time of maximum BC that increased LHRproxy occurs preferentially with large BC (Figure 5a). The occurrence distribution in the two-dimensional space spanned by BC (x-axis) and LHRproxy (y-axis) in Figure 11a depicts the general correlation between BC and LHRproxy, and the bin-averaged value of DIV is evidently a strong function of both, BC and LHRproxy: The largest DIV occurs for large BC with large LHRproxy, whereas the smallest values of DIV occur when BC is negative and LHRproxy is relatively small or negative. This strong coupling between dry dynamics (BC) and moist processes (LHRproxy) during baroclinic growth has been noted in many previous studies and is consistent with the underlying assumptions of many moist-baroclinic instability theories (e.g., see references in introduction).

In a first simple attempt to disentangle these inherent correlations, we consider the linear relationship between DIV and BC for fixed values of LHRproxy, and DIV and LHRproxy for fixed values of BC. The linear correlation coefficients and the slopes of the linear best fit are depicted in Figure 11b as a function of the respective fixed term. Both, the correlation coefficients and the slopes, are mostly larger for BC than for LHRproxy. This simple statistical analysis confirms that ridge amplification by DIV is strongly coupled to the underlying baroclinic development, as signified here by BC. Variations of LHRproxy, when considered for all values of BC, are less well suited to describe linear variations in DIV. This simple perspective, however, should not be taken as evidence that upper-tropospheric divergence is predominantly due to secondary circulations associated with the balanced, dry dynamics of the growing baroclinic wave.

Next, we consider in some more detail the variations of DIV with LHRproxy for different ranges of BC values. Considering bin-averaged values of DIV and LHRproxy as a function of BC (Figure 11c) reveals two distinct "regimes", separated by $BC = 0$: Both, $\partial DIV/\partial BC$ and $\partial LHRproxy/\partial BC$, are on average much larger for $BC > 0$ than for $BC < 0$. A further important characteristic of the two "regimes" is revealed when considering bin-averaged values of DIV and BC as a function of LHRproxy (Figure 11d): For $BC < 0$, there is no discernable systematic relationship between LHRproxy and DIV (on average, $\partial DIV/\partial LHRproxy \approx 0$) whereas for $BC > 0$ $\partial DIV/\partial LHRproxy$ is positive and large. This observation is most notable because $\partial BC/\partial LHRproxy$ is approximately constant[7]. This result based on a large number of real-world cases is consistent with expectations from moist-baroclinic instability theories: i) Figure 11c indicates the above-cited strong coupling between dry dynamics and moist processes during baroclinic growth. ii) Figure 11d indicates that the efficiency by which latent heat release leads to ridge building by divergent outflow depends on the underlying baroclinic development (BC). Baroclinic growth crucially depends on the phase relation between the upper- and lower-level PV anomalies, specifically for ridge amplification: between the warm sector of a cyclone, in which strong latent heat release within warm conveyor belts preferentially occurs, and the location of the ridge. Our interpretation is that BC here contains this phasing information, i.e., information on the

---

[7]Note that the functional dependence of bin-averaged values of LHRproxy on BC is not the inverse of the functional dependence of bin-averaged values of BC on LHRproxy (cf. Figure 11c and Figure 11d). The difference arises because the average value of LHRproxy of a specific BC bin is not equivalent to the value of the LHRproxy bin for that the average value of BC is approximately equal to the value of that specific BC bin.

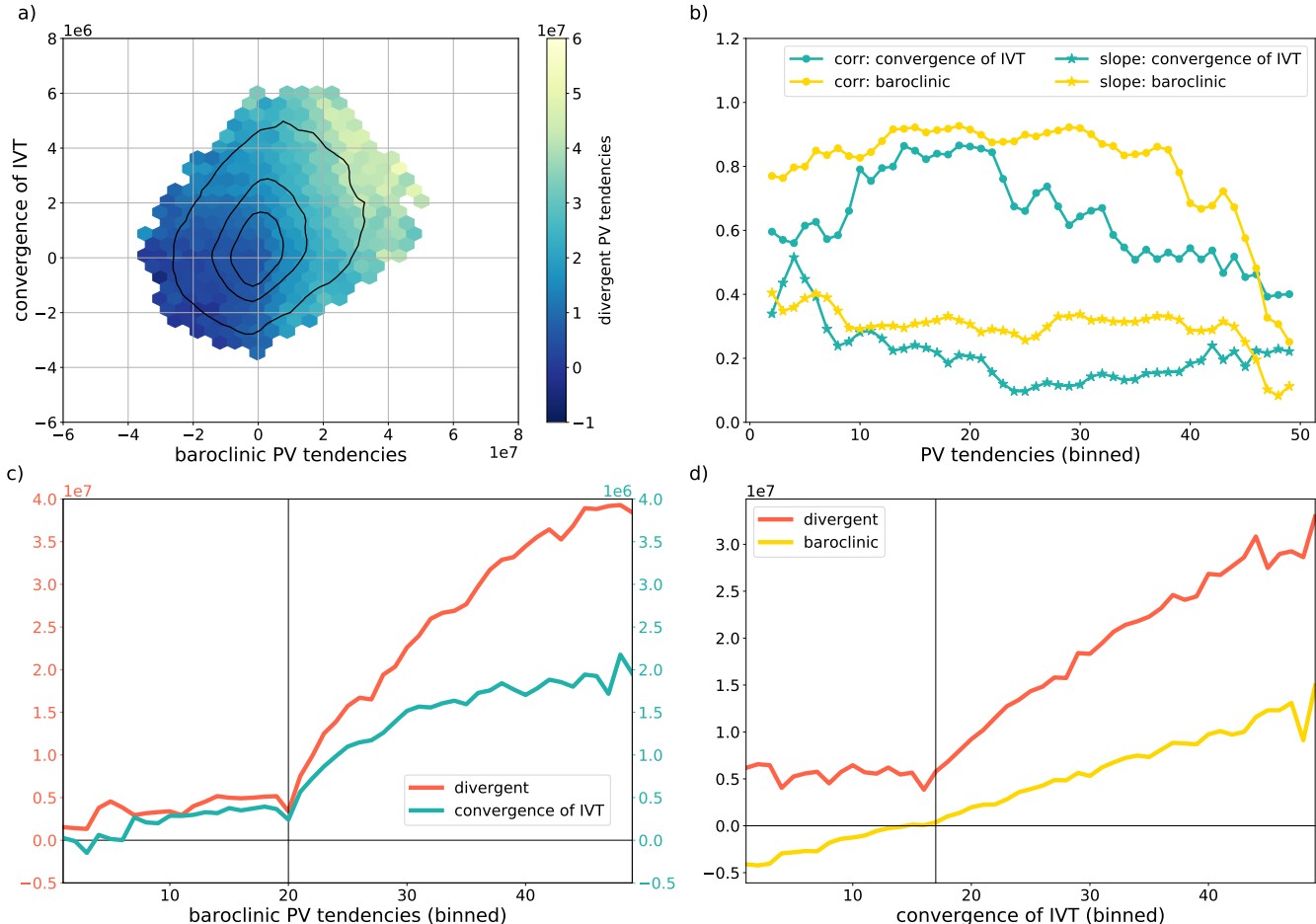

**Figure 11.** Relation between divergent (DIV) and baroclinic (BC) PV tendencies (in PVU m$^2$/s), and convergence of IVT (LHRproxy, in kg day$^{-1}$) within ridges. a) Binned scatter plot of BC (x-axis) and LHRproxy (y-axis) with average DIV shaded. Occurrence distribution shown by black contours (100,500,1000). b) Correlation (dots) and slope (stars) between (green) LHRproxy and DIV in each bin of BC and between (yellow) BC and DIV in each bin of LHRproxy. Values have been normalized by their respective standard deviation before regression. c) DIV (left y-axis) and LHRproxy (right y-axis) as function of BC. The vertical line depicts the bin for that BC turns positive. d) DIV and BC as function of LHRproxy. The vertical line depicts the bin for that LHRproxy turns positive. In b), c) and d) data divided into equally spaced bins from the 0.1% percentile to the 99% percentile, respectively.

relative position of latent heat and the upper-level ridge. Few moist-baroclinic instability theories (Mak, 1994; de Vries et al., 2010) consider the phasing of latent heat release explicitly. More commonly, the strong coupling of moist processes to the dry dynamics inherent in the theories implies that latent heat release invigorates the ascent associated with dry dynamics (e.g., Emanuel et al., 1987) and thus latent heat release is most effective for moist-baroclinic growth when the phasing for dry baroclinic growth is most favorable. Specifically for ridge amplification by divergent outflow, the importance of favorable

phasing has been emphasized in the context of the extratropical transition of tropical cyclones. In that context, the impact of latent heat release does not only depend on the magnitude of latent heat release but at least equally importantly on the relative position of latent heat release and the upper-tropospheric Rossby wave pattern (Keller et al., 2019; Riboldi et al., 2019). Our examination of a large number of real-world cases indicates that this notion of favorable phasing transfers to the more general case of divergent ridge amplification within RWPs.

We further examine the phasing aspect and the relation of DIV with BC and LHRproxy by considering spatial composites (Figure 12). Here, again, we approximately fix either BC or LHRproxy and then examine the spatial pattern that is associated with large variations of the other term. To fix one term, we only consider values of this term that are close to the median, specifically values within the 40-60% percentiles. As above, we create spatial composites centered on the ridge anomaly, but now only for values of one term within this near-median range and for values of the other term that exceed the 80% percentile

and fall below the 20% percentile, respectively. The design of these composites takes into account some of the variability of co-occurrence of LHR and BC, and thus moves beyond the strong coupling of moist processes and dry dynamics inherent in moist-baroclinic theories. The spatial composite of near-median values of BC and large values of LHRproxy are depicted in Figure 12a, which can be compared to the respective composite of small values of LHRproxy (Figure 12c). Both scenarios are characterized by a similar pattern of upper-level PV anomalies, in which the upstream and downstream troughs are of similar

amplitude and BC is small. In addition, the pattern of DIV is similar also. It is clear that the differences between these two scenarios are predominantly found in the magnitude of LHRproxy near the center of the ridge anomaly (near $(0°, 0°)$) and in the magnitude of DIV in the same region (Figure 12e). Figure 12e thus strongly indicates that differences in divergent ridge building in these scenarios can be attributed to differences in the amount of latent hear release.

The situation is more complex for the composites with near-median LHRproxy and with large and small values of BC,

respectively. By construction there is a large difference in BC, which extends over the whole ridge area (Figure 12f). Consistent with Figure 5, large BC occurs towards the trailing edge of the RWP (Figure 12b) whereas small BC occurs towards the leading edge (Figure 12d). These scenarios thus imply substantial differences of the upper-level PV anomalies, including differences in the shape of the ridge anomaly (Figure 12f): Ridges with large BC extend on average farther poleward on the upstream side than ridges with small BC. Importantly, the composites also exhibit differences in the pattern of LHRproxy. The maximum of

LHRproxy is located on average near the center of ridges with large BC (Figure 12b) but shifted polewards and downstream in ridges with small BC (Figure 12d). Figures (Figure 12b,d,f thereby illustrate differences in the phasing of latent heat release and the upper-level ridge anomaly that occur on average at two different stages of baroclinic development in a large number of real-world cases. Besides these phasing differences, maximum values of LHRproxy are larger with large BC than with small BC. By our definition of LHRproxy as a spatially integrated metric, these positive values are partly compensated by negative

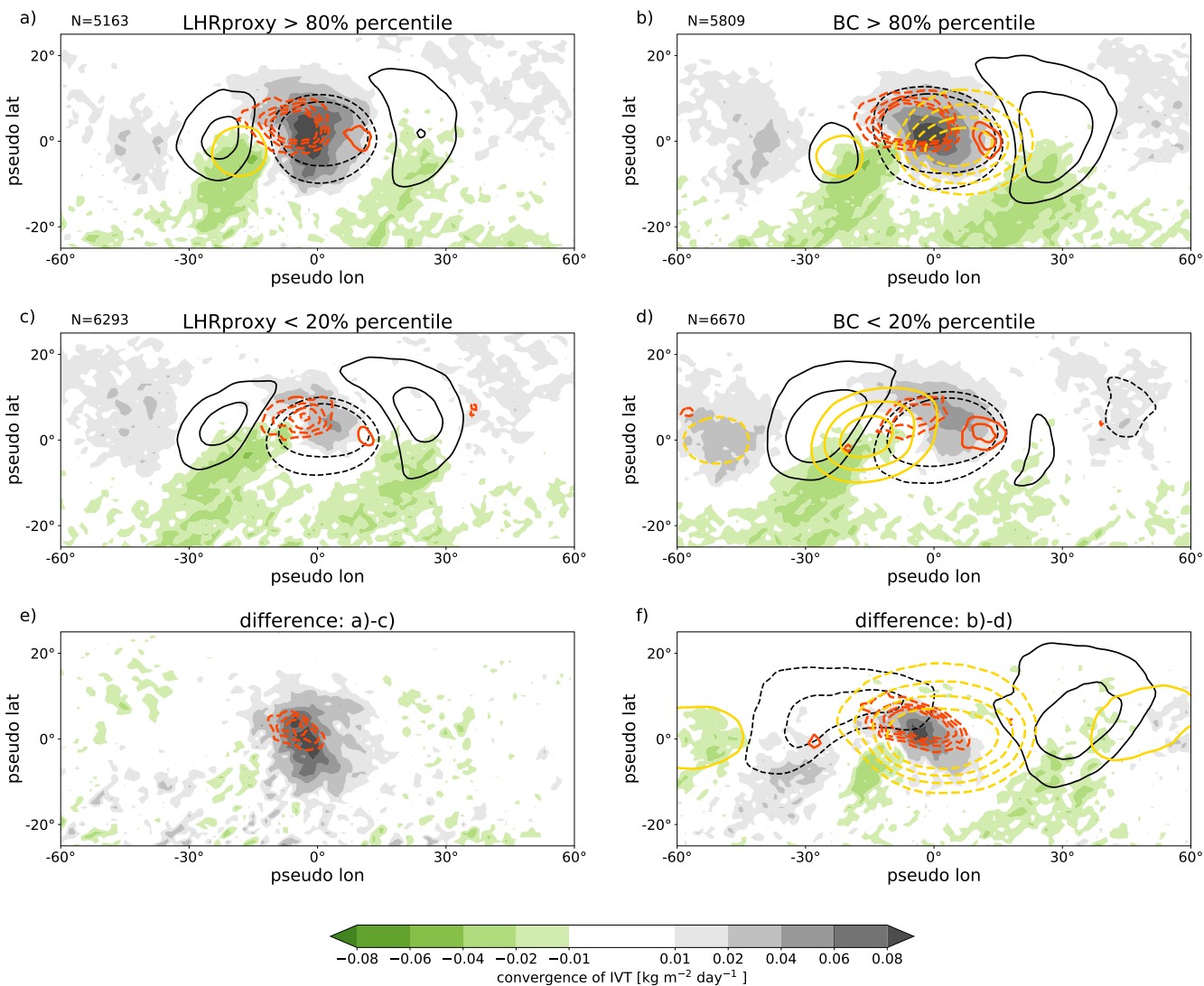

**Figure 12.** Spatial ridge-composites of convergence of IVT (LHRproxy, shading), and divergent (DIV) and baroclinic (BC) PV tendencies. Composites with average BC contribution (a,c,e) subdivided into a) strong LHRproxy (> 80% percentile), c) weak LHRproxy (<20% percentile) and e) difference plot. Composites with average LHRproxy (b,d,f) subdivided into b) strong BC (> 80% percentile), d) weak BC (<20% percentile) and f) difference plot. The different contours show in black PV anomalies ($\pm$(0.5, 1) PVU) , in red PV tendencies due to divergent flow ($\pm$(0.4, 0.6, 0.8, 1) PVU/day) and in yellow PV tendencies due to baroclinic interaction ($\pm$(0.2, 0.3, 0.4, 0.5) PVU/day). Negative tendencies dashed.

values on the upstream and equatorward side of the ridge, which are more strongly negative with large BC than with small BC (Figure 12f). Arguably, these negative values are associated with subsidence in the cold sector of a cyclone and positive values near the ridge center are associated with ascent in the warm sector (subsection 4.1), further illustrating the strong coupling of dry dynamics and moist processes during moist-baroclinic development. Notably, increased DIV for large BC coincides with the increased LHRproxy near the center of the ridge (Figure 12f)[8]. This striking coincidence strongly suggests that differences
in divergent ridge amplification can be attributed to differences in the release of latent heat below in these scenarios also.

Our analysis of the scenarios with near-median BC and large variations of LHRproxy and with near-median LHRproxy and large variations of BC (Figure 12) provides some evidence that divergent ridge building can predominantly be attributed to divergent outflow associated with latent heat release below. The simple statistical analysis presented in Figure 11b may not provide similar evidence because that analysis does not account for changes in the pattern and phasing of latent heat release that
occur on average during the baroclinic development. It is important to note, however, that the differences in DIV in Figure 12f are located also within the broad difference found in BC. The differences in DIV are further associated with differences in the amplitude and orientation of the upstream trough. We can thus not rule out that differences in the divergent flow that is part of secondary circulations associated with dry dynamics make further contributions to differences in divergent ridge amplification.

## 6   Conclusions

We have investigated the dynamics of troughs and ridges within RWPs in the northern hemisphere during the ERA5 period (1979-2017) in a quantitative, piecewise PV tendency framework. A comprehensive average picture of the dynamics is presented, extending and complementing previous analyses covering substantially shorter time periods. Compared to previous diagnostic frameworks, the PV perspective arguably provides a sharper view on the impact of nonconservative processes on the dynamics, most notably the impact of moist processes. The role of the divergent flow, by which moist processes most
prominently impact the evolution of RWPs, is explicitly accounted for in this framework. A caveat of our PV framework is that deformation, which plays a key role during wave breaking, is not accounted for. This caveat needs to be kept in mind in particular when interpreting the results during the late stage of ridge and trough life cycles.

Our results provide a quantitative confirmation of a pre-existing conceptual model of RWP dynamics (Fig. 9 of Wirth et al., 2018). On average, the quasi-barotropic near-tropopause dynamics are consistent with linear Rossby wave theory, leading to
660 amplification of anomalies towards the leading edge and weakening of anomalies towards the trailing edge of RWPs. Baroclinic and divergent amplification occur preferentially near the center and towards the trailing edge. Refining the conceptual model, our results show that i) the maximum divergent amplification of ridges occurs somewhat earlier during the life cycle than the maximum baroclinic growth and ii) baroclinic interaction on average weakens anomalies towards the leading edge.

---

[8]We have verified that all features discussed in this paragraph are indeed related to LHRproxy within the ridge area and not to LHRproxy within the neighbouring troughs. In addition, using a more sophisticated method to create the composites that enforces that the mean LHRproxy is the same in both composites (within 0.1%) yields virtually the same result. Between the presented composites, the mean LHRproxy differs by 4%

Baroclinic growth in summer is much weaker than in winter (about 50%). In contrast to winter, little relation between baroclinic growth and the overall amplitude evolution is found in summer when composites are centered on the time of maximum trough or ridge amplitude. For RWPs in summer, this observation may question the validity of the paradigm of downstream baroclinic development (Orlanski and Sheldon, 1995; Chang, 2000), which is well established and here confirmed for RWPs in winter. Focusing specifically on the sequence of governing processes by creating composites centered on the time of maximum baroclinic growth, however, clearly reveals that the paradigm does provide a valid description for northern hemispheric RWPs in summer, too.

The paradigm of downstream baroclinic development does not explicitly consider nonconservative processes, in particular latent heat release. Nonconservative tendencies from the YOTC data (available from May 2008 - April 2010) have been investigated to assess the impact of direct diabatic PV modification. Tendencies from parameterization schemes of longwave radiation, convection, clouds, and turbulence and orographic drag have been considered. The impact of the nonconservative tendencies is clearly dominated by longwave radiative cooling, which is comparable in magnitude to the advective tendencies. This impact is largely due to cross-isentropic transport of PV. The radiative tendency strengthens troughs and weakens ridges. The majority of these tendencies are associated with the climatological feature of a strong moisture gradient across the tropopause. A further notable part of the tendencies, however, can be associated with clouds and thus our results indicate the potential that cloud-radiative effects impact baroclinic development by the direct diabatic modification of upper-level PV anomalies. Further research is needed to explore how these cloud-radiative effects differ between individual cases and how they relate to cloud-radiative feedbacks on extratropical storm tracks. Tendencies due to latent heat release (the cloud and the convection scheme) exhibit a stronger link to the dynamical processes but are an order of magnitude smaller than the advective tendencies. Direct diabatic modification of upper-tropospheric PV by these processes thus has little impact on the overall evolution. Finally, tendencies due to turbulence and orographic drag have on average the smallest nonconservative impact and do not exhibit any notable signal in our analysis. Interestingly, the observed amplitude tendency of troughs and ridges during summer exhibits a small (about 10% relative amplitude) diurnal cycle. Our analysis indicates that this diurnal cycle arises due to a combination of the diurnal radiative (solar) cycle, and the diurnal convective cycle and its impact on upper-tropospheric divergence and radiative cooling.

While direct diabatic PV modification by latent heat release is small, moist processes have potentially a leading-order impact on RWPs by their indirect impact of invigorating upper-tropospheric divergence. Amplitude changes due to PV advection by the divergent flow are large and clearly related to the overall amplitude evolution. The divergent tendency consistently weakens troughs and amplifies ridges. The impact of the divergent flow is dominated by changes in the area of the anomalies, implying a shrinking of troughs and an extension of ridges. Differences in the spatial scale of troughs and ridges are a well known feature and can be explained to lowest order by upper-tropospheric divergence and convergence in (dry) semi-geostrophic theory (Hoskins, 1975; Wolf and Wirth, 2015), which accounts for the divergent flow to the extent that it is part of the ageostrophic wind. While dry theory explains the ridge-trough asymmetry to lowest order, we here show that moist processes, by invigorating upper-level divergence, contribute further to the observed asymmetry.

For ridges, the divergent tendency is strongly coupled to the baroclinic life cycle. In addition, the strongest ridge amplification is associated with an increased proxy of latent heat release within the ridge area. Our results thus provide further evidence

that divergent ridge amplification is closely coupled to moist-baroclinic development, confirming many previous studies that emphasize the role of latent heat release in warm conveyor belts for ridge building (e.g., Grams et al., 2011; Pfahl et al., 2015; Steinfeld and Pfahl, 2019). Consequently, the evolution of ridges is, on average, best described as downstream *moist*-baroclinic development.

While there is evidence that the impact of latent heat release is most prominently communicated to RWPs by upper-

tropospheric divergent outflow, it is in general difficult to accurately disentangle the relative contributions of dry and moist dynamics to upper-tropospheric divergence (e.g., Riemer et al., 2014; Quinting and Jones, 2016; Sanchez et al., 2020), and thus its impact on RWPs. This study does not attempt such an accurate quantitative decomposition. However, investigating scenarios with large variations of (a proxy for) latent heat release and baroclinic growth, respectively, while keeping the other process fixed at near-median values, does provide some evidence that divergent ridge building can predominantly be attributed

to divergent outflow associated with latent heat release below. In addition, our results demonstrate that divergent ridge amplification does not only depend on the magnitude of latent heat release but also on the location of latent heat release relative to the upper-level PV anomalies. As expected from theories of moist-baroclinic instability (e.g., Emanuel et al., 1987; Mak, 1994; de Vries et al., 2010), we observe that this phase relation becomes favorable when baroclinic growth commences and that it becomes increasingly more favorable when baroclinic growth increases. For real-world cases of divergent ridge amplification,

the importance of phasing has first been noted explicitly in the context of extratropical transition (Keller et al., 2019; Riboldi et al., 2019). Our examination of a large number of real-world cases indicates that the notion of favorable phasing transfers to the more general case of divergent ridge amplification within RWPs.

The current study has analyzed the mean dynamics of RWPs in the northern hemisphere. One avenue of future work is to perform a similar analysis for the southern hemisphere, where RWPs are less well organized in distinct storm tracks than in

the northern hemisphere. A further fruitful avenue is to analyze in more depth the variability of RWP dynamics, e.g., RWPs in the North Pacific and North Atlantic storm tracks, or RWPs in the context of other large-scale atmospheric features, such as ENSO, MJO, or the stratosphere. In addition, a future study will consider the dynamical differences between RWPs with high and low predictability, respectively, to gain insight into the important question under which conditions RWPs provide a large-scale source of enhanced predictability and under which conditions propagation and growth of forecast uncertainty

within RWPs lead to particularly low large-scale predictability. The mean perspective on RWP dynamics presented herein may provide a benchmark to identify anomalous dynamical behaviour in such more specific scenarios.

*Code and data availability.* ERA5 data from ECMWF can be downloaded from https://cds.climate.copernicus.eu/ and YOTC data from https://www.ecmwf.int/. The codes and data from this study can be provided by the authors upon request.

*Author contributions.* FT prepared the data, developed the computer algorithms, analyzed the data and created the figures. FT and MR formed the ideas and wrote the manuscript together.

*Competing interests.* The authors declare no competing interests.

*Acknowledgements.* The research leading to these results has been done within the subproject A8(N) of the Transregional Collaborative Research Center SFB / TRR 165 "Waves to Weather" (www.wavestoweather.de) funded by the DFG. In addition, the work was supported by the German Research Foundation (DFG) Grant RI 1771/4-1. We would like to thank Gabriel Wolf for providing his RWP catalogue and two anonymous reviewers whose comments helped to improve the presentation of our results.

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
