# Peer review of "Potential-Vorticity Dynamics of Troughs and Ridges within Rossby Wave Packets during a 40-year reanalysis period"

_Weather and Climate Dynamics, 2020_

## Referee Comment (RC1) · Anonymous Referee #1 · 10 Nov 2020

General Comments

The authors present a detailed study into the composite evolution of ridges and troughs within Rossby wave packets (RWPs), utilising a quantitative PV framework developed in previous publications. This is a well-studied problem but applying these diagnostics to it is certainly novel and has shed new light on some aspects of the dynamics involved, particularly with respect to the role of latent heating. They incorporate a large amount of data, by considering RWPs throughout the whole of ERA5, and consider the problem from several different complementary angles.

The manuscript is well written, and all figures are clear, and the results will certainly be

of interest to the wider community. I therefore recommend this paper is accepted for publication, subject to the following minor comments being addressed.

Specific Comments

L25: I'm not sure what you mean by the last sentence of the abstract. 'the most relevant aspect' in what respect?

L141: You call the first term on the RHS of Eq. 2 the 'adiabatic advection' of PV. This term is vague since, as you know, the wind field v is clearly modified by diabatic heating. I wonder if 'isentropic advection' provides a more accurate description? The term represents the advection of PV along isentropic surfaces (which makes a lot of sense when thinking about diabatic effects, see e.g. Harvey et al. (2020, QJRMS)), rather than the full 3-d material derivative following fluid parcels that many people are more used to thinking about, and 'isentropic advection' emphasizes this point. Also, I couldn't see where you defined v.

Sec. 3.2: It's commendable that you include all the details of the quality control you apply to your identified RWPs, and it's surely a complex task to filter out the events with 'questionable representativeness'. However, I was left wondering how you arrived at these thresholds. Have you tested the sensitivity of your results to any of these choices? In other words, how confident are you that you have succeeded?

Fig. 2 caption: Which axis is observed, and which is diagnosed? I may have misunderstood, but I wonder if 'amplitude tendency' is a better description of what is shown than 'amplitude evolution'? Also, what do you mean by '2d-fit', is it a least-squares regression? Finally, the symbol 'r' is often used for correlation, is there another symbol you can use for the slope here?

L276: 'weakening of ridges and an amplification of troughs' is confusing here because of the signs involved. Could you clarify whether you mean weakening of ridges or more negative PV tendencies, and how that relates to the offset from the origin in Figure 2.

L318: Just a comment. You note that the LHR is substantially stronger in winter than in summer, but that the divergent tendencies are similar. Are you able to tell why from your diagnostics? Is this because the divergent flow is similar in the two seasons, or because the PV gradients are weaker in summer than winter (or some other reason)? If the former, then is this just a result of having stronger static stability in winter?

L325: Could you expand on the methodology here. I think the composite time for each ridge/trough is based on the max/min values of the terms in Equation 6? Is that correct? Having just seen the spatial composites, I was not sure if it was that or some local maxima of the fields shown in Figure 3.

Figs 4, 5 and 6 captions: Using the words 'strongest' and 'weakest' could cause confusion here, due to anomalies taking both signs. Do you mean max and min? It might also help clarity if you reminded the reader that these plots include data from all seasons (in contrast to the Figure 3 which split into summer and winter), perhaps in the text at the start of section 4.2.

L385: I missed whether this section just uses the RWPs from the YOTC period, or all ERA5 RWPs with non-conservative tendencies only computed from the YOTC cases. Please could you clarify.

L436: I agree that the divergent flow has a detrimental impact on this measure of trough amplitude, based on area-integrated PV, but the mechanism is presumably much more adiabatic than the corresponding amplification of ridges, where mass is injection into the isentropic layer by the latent heating. I wonder if the depth-integrated mass-weighted PV [a more dynamically relevant measure of wave activity] also exhibits this effect?

L505: Again, just a comment. Is it obvious that divergence associated with the barotropic component does not also contribute to ridge building in the case of RWPs?

Technical Corrections

L74: 'occurrenc' -> 'occurrence'

L92: Should this read 'One prominent direct nonconservative impact'?

L139: This definition of \zeta_\theta is imprecise. Is it v_x - u_y with the derivatives evaluated along isentropic surfaces?

L324: 'at that the' -> 'at which the'

L356: 'baorclinic' -> 'baroclinic'

Fig 6 caption: You don't say what the arrows show, presumably the composite divergent wind?

L527: 'efficiency by that latent heat' -> 'efficiency by which that latent heat'

L563: 'the the' -> 'the'

---

## Referee Comment (RC2) · Anonymous Referee #2 · 25 Nov 2020

Review of WCD-2020-52:

"Potential-Vorticity Dynamics of Troughs and Ridges within Rossby Wave Packets during a 40-year reanalysis period" by F. Teubler and M. Riemer

General Comments:

Equations (1), (2) and (3) were first presented in this context by Davies and Didone (2013). However, there appear to be some sign mismatches to the derivation of Teubler and Riemer. In particular, the first term in (3) should have a minus in front and the same applies to the third and fourth term on the right-hand side. Even though the authors do not use these terms explicitly in large parts of their study, as they mainly evaluate the

adiabatic terms, these errors should be corrected if the equation is to be maintained in the manuscript.

The PV partitioning is not clear. Do the authors assign the entire potential temperature anomaly at the upper and lower boundary exclusively as boundary condition to the upper and lower PV anomaly, respectively, and for inversion a zero potential temperature anomaly is assumed at the other boundary? What is the justification for such an assignment? Or in other words, why should the upper level PV anomaly not significantly project onto the lower boundary and vice versus? Can the exclusion of such an influence be justified?

In section 4.2, the authors talk about "amplification", though it is not clear what they mean by that. If the advective tendencies that the authors discuss in this section should have an amplifying effect, there should be an alignment of the tendency with the actual anomaly. However, the phase shift between the tendency and the anomalies is more or less 90 degrees, which implies, as the authors pointed out in a previous section, a propagating response, not an amplifying response. This also renders the discussion about the relative weight of up- and downstream PV anomalies in this context questionable. In a way, the amplitude of the tendency is larger where it is located between larger PV anomalies, which is not surprising if this pattern is supposed to be propagated by advection. Based on these arguments, the conclusion of the authors in the second paragraph of this section about amplification associated with the quasi-barotropic advection is misleading.

Similar to the arguments in the previous general comment, the amplification associated with the baroclinic tendencies also needs some revision. As there is now significant alignment of the tendency with the PV anomaly, it is correct to refer to an amplification. However, the fact that the largest PV anomaly has the largest tendency does not mean that it is growing the fastest in a relative sense. For example, if a pattern would be growing exponentially, such as in the Eady model of baroclinic instability, and if there would be a smaller and larger anomaly, they might feature the identical exponential

growth, but the absolute tendency of amplification is different due to the different ampli-tudes. Therefore, the meaning of amplification needs to be clarified. For example, do the arguments of the authors hold if one diagnoses relative tendencies, i.e., normalized by the amplitude of the respective PV anomaly. It appears from Fig. 5 that this might be the case for some parts of the RWP but not in a general sense.

In section 5.1, the authors present a comparison between the dynamic and thermody-namic terms, where only radiation plays an appreciable role in the overall development, while latent heating and other terms are rather minor. However, when looking at Fig. 9, it appears that in the evolution of the packet, there is almost no variation in the con-tribution from radiation, as indicated by the authors in section 5.2. In general, it would be good if the authors could expand the discussion around these terms and put the dy-namic and thermodynamic contributions in better context. Furthermore, if my reading of the methodology is correct, the upper level PV anomaly and therefore its tendency, is defined between 600 and 150 hPa. Thus, the PV anomaly is defined across the tropopause interface between the troposphere and stratosphere, where a very strong vertical PV gradient and densely spaced potential temperature surfaces are present. In such a setup, the slightest heating will result in a strong response in PV, also from radi-ation. However, the relevance of these PV anomalies if they are across the tropopause is maybe not significant. Can the authors expand on where the respective heating occurs with respect to the PV gradients and the tropopause?

Another comment on the radiation results would be that there has been significant focus on the effects of radiation, in particular related to cloud tops, on storm tracks (e.g., work by Aiko Voigt and the Cookie experiment community), and thus implicitly on cyclones on RWPs. The results of the authors indicate that the claimed impact by the aforementioned community might not be as relevant on a feature-based view and mainly reflect itself in a climatological background, which would be worthwhile to put in context.

Related to the comment above, most likely, most heating associated with cloud and

rain processes occurs below 600 hPa and therefore the tendencies of upper level PV are not directly affected by the diabatic terms. However, the displacement of the theta surfaces will be felt aloft, which will manifest itself in divergence at these levels. For example, heating at levels below 600 hPa yields a mass transport above potential temperature surfaces that can be located below 600 hPa initially, but the ensuing mass redistribution will reach higher altitudes in the hydrostatic and geostrophic adjustment. The divergent signal is thus potentially largely associated with the atmosphere trying to attain balance after experiencing diabatic heating. The method employed by the authors cannot disentangle between these causes and effects. The authors should expand the discussion about these potential caveats and what they imply for the interpretation of the results.

In general, I found it sometimes difficult to follow the reasoning of the authors and they sometimes also indicate that they will contradict themselves, e.g., paragraph 508-519, especially line 518. As a reader, I would appreciate a more stringent guidance through the material avoiding potential confusions and suspense to wait (at the cost that I might have forgotten until then) until further clarification in later sections.

I find the argumentation in the ensuing paragraph also confusing (top of page 25). The authors make it sound like as if the moist baroclinic paradigm does not require a beneficial phasing of the latent heating with the overall baroclinic structure. This would be incorrect and indeed the arguments presented by the authors at the end of the paragraph are consistent with the moist-baroclinic instability paradigm, i.e., the phase relation between heating and the perturbations is closely tied to the overall baroclinic structure. Furthermore, most of what is then argued in the last paragraph before the conclusions is straight forward moist-baroclinic instability reasoning, i.e., once there is a baroclinic growth, i.e., once a westward vertical tilt of the anomalies is established in a baroclinic environment, ascent and associated latent heating occurs at a favorable location for growth. The authors thus do not present something new in this context, even though they appear to make it seem like new. Instead of presenting these findings

as something new, they should rather put their findings in context with existing literature on moist baroclinic instability and relate their findings to the overall three-dimensional structure of the RWP.

Specific Comments: Reference to line numbers in the manuscript.

163: Has this approximate equality been checked with data? I am wondering how close this relation really holds.

170-171: The authors state that the aforementioned occurrences are exceptional. Why do the authors not identify and quantify the potential influence of these occurrences and the effect mid-level PV anomalies might have on their results?

259: There is a grammar issue with the sentence, in particular with the run-on part "and mainly due to small-scale...".

297-298: How can one infer the group velocity from the blue contours? Strictly speaking, given the 90 degrees shift, one can only identify the phase propagation.

377: "save"? What does that mean in this context?

399: The value is not "random", as the authors state it is reflecting a more "climatological" value.

560: How can adiabatic subsidence be of significance in an isentropic framework? It would basically be invisible, except if it is associated with horizontal divergence.

614: "tropospheric"

637: Please clarify what is meant by "differences in phasing" in this context. Phasing of what? Furthermore, what aspects of the secondary circulation are meant?
* * *

---

## Author Response (AR1)

**Reply to the Reviewers**

**Reviewer #1**

**General Comments**

The authors present a detailed study into the composite evolution of ridges and troughs within Rossby wave packets (RWPs), utilising a quantitative PV framework developed in previous publications. This is a well-studied problem but applying these diagnostics to it is certainly novel and has shed new light on some aspects of the dynamics involved, particularly with respect to the role of latent heating. They incorporate a large amount of data, by considering RWPs throughout the whole of ERA5, and consider the problem from several different complementary angles.

The manuscript is well written, and all figures are clear, and the results will certainly be of interest to the wider community. I therefore recommend this paper is accepted for publication, subject to the following minor comments being addressed.

We thank the reviewer for her/ his insightful comments that helped to further improve our manuscript. Our responses to the comments are given below.

**Specific Comments**

L25: I'm not sure what you mean by the last sentence of the abstract. 'the most relevant aspect' in what respect?

Upon reflection, we agree that this last sentence is hard to understand, in particular before having read the manuscript. The sentence refers to the role of dry dynamics in the divergent amplification of ridges considered in some detail in our Sect. 5.3. Based on the comment of another reviewer, we have revised that section to some extend and have clarified the wording of the last sentence in the abstract.as follows: "...we provide some evidence that variability in the strength of divergent ridge amplification can predominantly be attributed to variability in latent heat release below, rather than to secondary circulations associated with the dry dynamics of a baroclinic wave."

L141: You call the first term on the RHS of Eq. 2 the 'adiabatic advection' of PV. This term is vague since, as you know, the wind field v is clearly modified by diabatic heating. I wonder if 'isentropic advection' provides a more accurate description? The term represents the advection of PV along isentropic surfaces (which makes a lot of sense when thinking about diabatic effects, see e.g. Harvey et al. (2020, QJRMS)), rather than the full 3-d material derivative following fluid parcels that many people are more used to thinking about, and 'isentropic advection' emphasizes this point. Also, I couldn't see where you defined v.

We agree and have renamed the term to isentropic advection. In addition, v is now defined.

Sec. 3.2: It's commendable that you include all the details of the quality control you apply to your identified RWPs, and it's surely a complex task to filter out the events with 'questionable representativeness'. However, I was left wondering how you arrived at these thresholds. Have you tested the sensitivity of your results to any of these choices? In other words, how confident are you that you have succeeded?

We have tested the sensitivity to the threshold for the differences between the observed and the diagnosed tendencies, for which we have eventually applied the 1.5-IQR rule, in some detail. The results of these tests are described in Sect. 3.3. Based on your comment, we note that this description is not sufficiently clear. What our sensitivity test showed is that the bias in the slope of the diagnosed tendencies in Fig. 2 stems from missing amplification by merging

and weakening by splitting events: Changing the threshold for the IQR rule mostly changed the distribution shown in Fig. 2 for large absolute values of observed values. The mean values of diagnosed tendencies lie very close the diagonal for small and moderate absolute values, i.e., for the vast majority of data, irrespective of the choice of threshold. We are thus confident that the mean-picture presented in this study is not contaminated by data of questionable representativeness. Understanding that the bias observed in Fig. 2 is arguably due to missing partial merging and splitting events, we chose to be not particularly restrictive with the threshold for the IQR rule.

We have modified the manuscript to reflect this information.

Using the 3-IQR rule in the cases described in Sect. 3.2 affects few data and, according to literature, 3-IQR is a standard value for eliminating outliers. We thus spent less time testing sensitivities with respect to the choice of this threshold. Using values of 2 and 4 did not change the results shown in Fig. 7 in any notable way. We have added the information that the 3-IQR rule is a standard choice.

Fig. 2 caption: Which axis is observed, and which is diagnosed? I may have misunderstood, but I wonder if 'amplitude tendency' is a better description of what is shown than 'amplitude evolution'? Also, what do you mean by '2d-fit', is it a least-squares regression? Finally, the symbol 'r' is often used for correlation, is there another symbol you can use for the slope here?

Thanks for pointing this out. We have added the axis labels. In addition, we agree that 'amplitude tendency' is a more precise term in this context and have changed the text accordingly. With 2d-fit we wanted to point out that we use total least squares instead of ordinary least squares to perform linear regression. As you might recall ordinary least squares only try to minimize the residual between the y-axis variable and the fit and do not account for uncertainties in the x-axis variable. Since our x-axis variable contains also uncertainties, we try to minimize the residual of both variables with the fit. We now clarified this in the caption of Fig.2 and changed the symbol r.

L276: 'weakening of ridges and an amplification of troughs' is confusing here because of the signs involved. Could you clarify whether you mean weakening of ridges or more negative PV tendencies, and how that relates to the offset from the origin in Figure 2.

We agree. This description needs clarification. More confusing, however, is the fact that we accidentally had omitted the important information that the tendencies for the ridges in Fig. 2 had been multiplied by (-1) such that positive tendencies would denote amplification for both, ridges and troughs. We apologize for this omission, which has most likely contributed to the confusion. We now prefer to show the figure without this rather confusing modification. In addition, we have modified the text and now write for increased clarity "… weakening of ridge amplitude and an amplification of trough amplitude." in the last paragraph of Sect. 3.3.

L318: Just a comment. You note that the LHR is substantially stronger in winter than in summer, but that the divergent tendencies are similar. Are you able to tell why from your diagnostics? Is this because the divergent flow is similar in the two seasons, or because the PV gradients are weaker in summer than winter (or some other reason)? If the former, then is this just a result of having stronger static stability in winter?

Before submission, we had thought about this rather curious observation, too. Unfortunately, we could not find a non-speculative explanation. Differences in static stability could be one explanation but our diagnostic does not provide a straight-forward means to test the idea. With a weaker PV gradient (in summer) we would expect weaker PV tendencies from the same upper-tropospheric divergence so this is likely not an explanation. A further potential explanation is that LHR in summer is more often associated with convection, which is potentially not sufficiently resolved by our proxy of LHR, whereas in winter LHR is mostly "on

the grid scale". Again, we did not find a straight-forward way to test this potential explanation. In the manuscript, we prefer to refrain from speculative explanations.

L325: Could you expand on the methodology here. I think the composite time for each ridge/trough is based on the max/min values of the terms in Equation 6? Is that correct? Having just seen the spatial composites, I was not sure if it was that or some local maxima of the fields shown in Figure 3.

Yes, you are right. It is based on the terms in Equation 6. We now made this point clear at the beginning of Section 4.2.

Figs 4, 5 and 6 captions: Using the words 'strongest' and 'weakest' could cause con- fusion here, due to anomalies taking both signs. Do you mean max and min? It might also help clarity if you reminded the reader that these plots include data from all seasons (in contrast to the Figure 3 which split into summer and winter), perhaps in the text at the start of section 4.2.

Thank you, we have added a reminder at the start of Sect. 4.2.

For clarity, we have changed the caption to "... for ridges (a,c) and troughs (b,d) at the times when the quasi-barotropic PV tendencies yield maximum amplification (a,b) and maximum weakening (c,d) of the respective amplitude."

L385: I missed whether this section just uses the RWPs from the YOTC period, or all ERA5 RWPs with non-conservative tendencies only computed from the YOTC cases. Please could you clarify.

The latter. We clarified this point in the text.

L436: I agree that the divergent flow has a detrimental impact on this measure of trough amplitude, based on area-integrated PV, but the mechanism is presumably much more adiabatic than the corresponding amplification of ridges, where mass is injection into the isentropic layer by the latent heating. I wonder if the depth-integrated mass-weighted PV [a more dynamically relevant measure of wave activity] also exhibits this effect?

We thank the reviewer in particular for this comment. In our framework, the impact of the divergent wind on the area-integrated PV anomaly is two-fold (Eq. 6 in the manuscript): i) advection of background PV and ii) change in the area of the PV anomaly. Considering the second effect in isolation, the horizontal boundaries of an anomaly constitute material surfaces. Considering the anomaly between two isentropic levels results in a material volume for adiabatic motion. According to the impermeability theorem (Haynes and McIntyre 1987,1990), the density-weighted PV (or "PV substance", to which we believe that the reviewer refers) will not change for adiabatic motion. An amplitude metric defined based on such a volume integral of PV substance should thus not yield an amplitude change due to this second effect of the divergent wind. As noted by the reviewer, the situation is different if diabatic transport effectively changes the mass of a PV anomaly sandwiched between two isentropic levels (as in the case of ridge building as indicated by our results).

We appreciate that the reviewer points to this intriguing between PV-based amplitude metrics. Because there are no other references to PV substance or wave activity in our manuscript, we prefer not to point out this interesting aspect in this study.

L505: Again, just a comment. Is it obvious that divergence associated with the barotropic component does not also contribute to ridge building in the case of RWPs?

Thank you for noting this unclarity. Our premise here is that the dry component of upper-tropospheric divergence varies to lowest order with the stage of the baroclinic life cycle. With the available PV tendencies as proxies, the stage of the baroclinic life cycle is arguably more

closely related to BC than to the barotropic component. Indeed, this reasoning is currently not expressed sufficiently clear and we have clarified it in the revised version.

**Technical Corrections**

Thank you for the careful reading. We have corrected the manuscript accordingly.

L74: 'occurrenc' -> ,occurrence'thanks

L92: Should this read 'One prominent direct nonconservative impact'?

Thank you, our wording here was indeed somewhat unclear. We meant to say: "One prominent indirect nonconservative impact are advective tendencies by the winds associated with low-level PV anomalies generated by latent heat release, in particular their role in enhancing baroclinic growth." We have changed the manuscript accordingly.

L139: This definition of $\zeta_\theta$ is imprecise. Is it $v_x - u_y$ with the derivatives evaluated along isentropic surfaces? Yes. For clarification we now write "... the component of relative vorticity perpendicular to an isentropic surface."

L324: 'at that the' -> 'at which the' thanks

L356: 'baorclinic' -> 'baroclinic' thanks

Fig 6 caption: You don't say what the arrows show, presumably the composite divergent wind? Thanks, yes, we have added this information to the caption of Fig. 6.

L527: 'efficiency by that latent heat' -> 'efficiency by which that latent heat' thanks

L563: 'the the' -> 'the' thanks

**Reviewer #2**

We thank the reviewer for her/ his careful reading of our manuscript and the thought-provoking comments. The comments help to further improve our manuscript. Our responses to the comments are given below.

**General Comments:**

1. Equations (1), (2) and (3) were first presented in this context by Davies and Didone (2013). However, there appear to be some sign mismatches to the derivation of Teubler and Riemer. In particular, the first term in (3) should have a minus in front and the same applies to the third and fourth term on the right-hand side. Even though the authors do not use these terms explicitly in large parts of their study, as they mainly evaluate the adiabatic terms, these errors should be corrected if the equation is to be maintained in the manuscript.

Thank you for carefully checking the equations. Indeed, there is a typo and the first term on the right-hand side of equation (3) should have the opposite sign. We have double checked our code and have confirmed that our implementation is correct.

Regarding the third term, we agree that splitting the diabatic term in a "stretching" (second term) and "tilting" (third term) contribution is non-standard and that we have adopted this formulation from Davies and Didone (2013). We now choose to present the "standard" form of the nonconservative impact on PV in isentropic coordinates, i.e., we omit the splitting because we do not consider the individual contributions of diabatic heating in this manuscript.

The signs of the third and the fourth term in our original manuscript are correct. For the fourth term It is clear that relative vorticity and thus PV increases if the curl of the accelerations (v

dot) is positive. The sign of the third term can be verified rather easily by explicit calculation of the splitting of the diabatic term. Note that there are incorrect signs in Eq. 3a, 3b, and 4 in Davies and Didone (2013)) and in Eq. 72 and 73 in Hoskins et al. (1985), which may cause quite some confusion when trying to verify the equations!

2. The PV partitioning is not clear. Do the authors assign the entire potential temperature anomaly at the upper and lower boundary exclusively as boundary condition to the upper and lower PV anomaly, respectively, and for inversion a zero potential temperature anomaly is assumed at the other boundary? What is the justification for such an assignment? Or in other words, why should the upper level PV anomaly not significantly project onto the lower boundary and vice versus? Can the exclusion of such an influence be justified?

Our piecewise PV inversion inverts i) the low-level PV anomalies between 850 and 650hPa) together with the temperature anomalies at 875hPa and ii) the upper-level PV anomalies between 600hPa and 150hPa together with the temperature anomalies at 125hPa. For the upper-level inversion we thus do not assume any anomalies (PV and temperature) below 600hPa, and vice versa for the inversion of low-level PV anomalies. The idea to consider boundary theta anomalies as distinct anomalies traces back to the Eady model. The role of theta anomalies as (delta-distributed) PV anomalies has been made explicit by Bretherton (1966). This is at the heart of PV partitioning and the idea of counter-propagating Rossby waves (e.g., references in manuscript: Hoskins et al. 1985, Emanuel et a. 1987, Heifetz et al. 2004b, de Vries et al. 2009). Such a partitioning was hence used in many previous studies employing piecewise PV diagnostic (e.g., Davis and Emanuel 1991, Davis et al. 1996, Nielsen-Gammon and Lefevre 1996, and our own previous work). Due to this standard use of the partitioning, we do not agree that the partitioning is unclear, at least as long as one accepts the standard paradigm of PV partitioning.

If we understand correctly, the reviewer challenges this paradigm and is concerned that static stability anomalies associated with the upper-level PV anomalies could penetrate down to the low-level boundary and imprint on the boundary theta distribution. We commend the reviewer on this out-of-the-box thinking and appreciate this thought-provoking question. In principle the reviewer is correct that the upper-level anomalies may impact the low-level theta distribution by associated stability anomalies. Stability anomalies arise because of a vertical deflection of theta surfaces by adiabatic vertical motion, which, in turn arises as part of secondary circulations during an adjustment-to-balance process. If the lower boundary were defined as a rigid boundary, i.e., the Earth's surface, then there could not be any theta anomaly associated with the upper-level PV distribution because vertical motion vanishes at the rigid lower boundary. In "standard" PV thinking for real-world applications, however, the lower boundary is defined at the top of the planetary boundary layer to avoid "contamination" of the balanced dynamics by boundary-layer processes. Vertical motion thus does not need to vanish at a such-defined lower boundary and, in principle, the boundary theta distribution could be modified by upper-level PV anomalies. Vertical motion, and thus theta anomalies associated with upper-level PV anomalies, however, can be expected to be small at the top of the planetary boundary layer due to the closeness to the rigid bottom where vertical motion needs to vanish (in more technical terms: vertical motion associated with upper-level balanced (PV) dynamics can be solved for by a variant of an omega-equation, i.e., by inverting an elliptic partial differential equation. The boundary condition vertical motion = 0 will ensure that vertical motion approaches zero when approaching the boundary). In addition, there is a density effect, which dictates that vertical motion needs to decrease with increasing density, i.e., height to fulfill continuity (in the absence of horizontal motion). In summary, there are sound theoretical arguments why low-level theta anomalies associated with upper-level PV can expected to be small. Synoptic experience supports the theoretical considerations. In practice, the "standard" separation of PV anomalies is thus well justified, although in principle there may be a non-zero imprint of the upper-level PV anomalies on theta at the top of the boundary, i.e., the lower boundary used for piecewise PV inversion. Analogous arguments apply for the

upper boundary, where the high stability of the stratosphere effectively plays the role of the "rigid" boundary.

3. In section 4.2, the authors talk about "amplification", though it is not clear what they mean by that. If the advective tendencies that the authors discuss in this section should have an amplifying effect, there should be an alignment of the tendency with the actual anomaly. However, the phase shift between the tendency and the anomalies is more or less 90 degrees, which implies, as the authors pointed out in a previous section, a propagating response, not an amplifying response. This also renders the discussion about the relative weight of up- and downstream PV anomalies in this context questionable. In a way, the amplitude of the tendency is larger where it is located between larger PV anomalies, which is not surprising if this pattern is supposed to be propagated by advection. Based on these arguments, the conclusion of the authors in the second paragraph of this section about amplification associated with the quasi-barotropic advection is misleading.

Our definition of the tendencies that govern "amplitude" evolution are given in some detail in Sect. 2.3. Our amplitude metric is the PV anomaly spatially integrated over the region of the anomaly. Based on the reviewer's comment, we realize that this definition may not be explicit enough and have modified the text to clarify our use of the term amplitude. The reviewer is correct that one prominent signal in the quasi-barotropic tendencies is a 90 degree phase shift. A further signal, which we explicitly describe in the second paragraph of Sect. 4.2, is that amplifying tendencies dominate over the weakening tendencies. The reviewer's comment makes it clear to us that we should have noted here explicitly that this distribution of tendencies leads to amplification in the spatially integrated sense that is considered in our study. We now do so in the revised manuscript. The discussion of the relative weight of anomalies and our conclusions are thus not in question.

4. Similar to the arguments in the previous general comment, the amplification associated with the baroclinic tendencies also needs some revision. As there is now significant alignment of the tendency with the PV anomaly, it is correct to refer to an amplification. However, the fact that the largest PV anomaly has the largest tendency does not mean that it is growing the fastest in a relative sense. For example, if a pattern would be growing exponentially, such as in the Eady model of baroclinic instability, and if there would be a smaller and larger anomaly, they might feature the identical exponential growth, but the absolute tendency of amplification is different due to the different amplitudes. Therefore, the meaning of amplification needs to be clarified. For example, do the arguments of the authors hold if one diagnoses relative tendencies, i.e., normalized by the amplitude of the respective PV anomaly. It appears from Fig. 5 that this might be the case for some parts of the RWP but not in a general sense.

We agree with the reviewer that a note may be helpful to avoid confusion of readers that may expect an analysis of growth rates at this point. We have added to the manuscript: "Note that we here consider the absolute growth of anomalies rather than growth rates, which are often considered in more theoretical studies of baroclinic instability."

In general, a different choice of the definition of "growth" may yield somewhat different results. We deliberately choose to examine absolute growth and not relative growth of anomalies. One reason for this choice is that small anomalies during the initial stage of their development are hard to reliably identify, at least with our identification and tracking method. To avoid introducing an associated bias to relative growth rates, we prefer to study the absolute growth of anomalies.

5. In section 5.1, the authors present a comparison between the dynamic and thermodynamic terms, where only radiation plays an appreciable role in the overall development, while latent heating and other terms are rather minor. However, when looking at Fig. 9, it appears that in the evolution of the packet, there is almost no variation in the contribution from radiation, as indicated by the authors in section 5.2. In general, it would be good if the authors could expand

the discussion around these terms and put the dynamic and thermodynamic contributions in better context. Furthermore, if my reading of the methodology is correct, the upper level PV anomaly and therefore its tendency, is defined between 600 and 150 hPa. Thus, the PV anomaly is defined across the tropopause interface between the troposphere and stratosphere, where a very strong vertical PV gradient and densely spaced potential temperature surfaces are present. In such a setup, the slightest heating will result in a strong response in PV, also from radiation. However, the relevance of these PV anomalies if they are across the tropopause is maybe not significant. Can the authors expand on where the respective heating occurs with respect to the PV gradients and the tropopause?

Thank you for this comment. Originally, we believed that a further discussion of the radiative tendencies is beyond the scope of our study. Based on this comment, however, we realize that some more information on the radiative tendencies is of interest here and will improve interpretation and clarity. We have extended  our manuscript in this regard. First, we refer to previous work (Zierl and Wirth 1997, Chagnon et al 2013) when first discussing the general characteristic of longwave radiation to weaken ridges and to strengthen troughs in Sect. 5.1. In addition, we have clarified our reference to Gristey et al. (2018), which now reads: "Gristey et al. (2018) have shown that the impact of convection on radiation modifies the diurnal cycle of longwave radiation, which is predominantly governed by the diurnal cycle of land surface heating, and the associated emission temperature, by insolation."

More substantially, we have added a new figure (new Fig. 10) that depicts spatial composites of the radiative PV tendencies in horizontal and vertical cross sections, in comparison with a rough proxy for cloud occurrence. These composites show distinct extrema in radiative tendencies in clear association with our cloud proxy, thereby demonstrating a clear potential for cloud-radiative feedback. The composites allow to estimate the strength of the extrema relative to a background value. The discussion has thus substantially improved in response to your comment, Two paragraphs have been added at the end of Sect. 5.2 to describe and interpret the spatial pattern of the radiative tendencies. In addition, we have put our results in the context of the recent large interest in cloud-radiative effects on extratropical storm tracks (see also your next comment). The two new paragraphs read:

"The longwave radiative tendency is largely constant during the baroclinic life cycle of both, ridges and troughs. Because this tendency makes a contribution to the amplitude evolution comparable to the advective tendencies, and due to the large recent interest in cloud-radiative feedback on extratropical storm tracks and on cyclone evolution (e.g., Schäfer and Voigt 2018, Grise et al. 2019, Papavasileiou et al. 2020), we investigate the radiative tendency in some more detail. Figure 10 depicts the spatial pattern of this tendency (orange contours). To lowest order, the pattern can be considered to comprise two components. The first component is characterised by isolines that are parallel to the undulated tropopause, with values increasing from the troposphere to the stratosphere. This increase is arguably associated with the strong moisture gradient across the tropopause. Superimposed on this "background" component are local maxima ahead of troughs with values that are approximately 50\% larger than the "background" values. The location of these maxima is consistent with a schematic of the typical radiative impact on the synoptic-scale wave pattern by Chagnon et al. (2013, their Fig. 10, hypothesized from the results of a detailed case study. The spatial pattern of the radiative tendencies thus reveals important variations, in contrast to the temporal evolution of the spatially-integrated values relative to the maximum of baroclinic growth (Figure 9). In particular, the maximum ahead of troughs can be associated with a maximum in the occurrence of clouds: Large-scale ascent and associated cloud formation is usually expected ahead of a trough. Using the PV tendencies from the cloud scheme as a rather rough, but from our data easily available proxy for cloud occurrence confirms the clear relation of clouds and the extrema in radiative PV tendencies (Figure 10). Importantly, the patterns and their amplitudes depicted in Figure 10 do not vary appreciably over the average baroclinic life cycle, as here defined from minus to plus three days around the maximum of baroclinic growth: At any stage, baroclinic development occurs within a wavy upper-level pattern with an extremum

of our proxy for cloud occurrence ahead of the trough (not shown). This continued existence of an upper-level wave pattern is consistent with our selection of individual cases as being part of RWPs.

Our interpretation of the results for the radiative tendencies is that the majority of the modification of trough and ridge amplitude is associated with "background" radiation, i.e., with radiative tendencies that are associated with the climatological feature of a strong moisture gradient across the tropopause. It seems plausible that this impact exhibits little coupling with the underlying dynamics (as found in Figure 9). A further, notable part of the radiative tendencies is apparently associated with cloud-radiative effects. There is thus the potential that cloud-radiative effects impact baroclinic development by the direct diabatic modification of upper-level PV anomalies. From the results of this study alone, however, it is not straight-forward to compare our findings to the existing literature on cloud-radiative feedback on storm tracks. Most importantly, cloud-radiative effects on extratropical storm tracks are more complex than by direct upper-level PV modification (Grise et al. 2019). The lack of variation with the baroclinic life cycle found herein hinges on the continued existence of a wavy upper-level pattern. Examinations of different scenarios, e.g., idealized life cycles that start from a straight jet may yield a different result. In addition, the lack of variation over the composite life cycle does not exclude the potential for important differences between individual cases, which warrants future investigations into the case-to-case variability of the cloud radiative component of direct diabatic PV modification of troughs and ridges."

In addition, we have revised related text in the abstract and the conclusions. The revised abstract now reads "Longwave radiative cooling makes a first-order contribution to ridge and trough amplitude, with the potential that this contribution is partly associated with cloud-radiative effects." (First part of the sentence remained unchanged. The word limit of the abstract prohibits a more complete reflection of the revised interpretation of the role of longwave radiation.) The revised conclusions now note that: "The majority of these tendencies are associated with the climatological feature of a strong moisture gradient across the tropopause. A further notable part of the tendencies, however, can be associated with clouds and thus our results indicate the potential that cloud-radiative effects impact baroclinic development by the direct diabatic modification of upper-level PV anomalies. Further research is needed to explore how these cloud-radiative effects differ between individual cases and how they relate to cloud-radiative feedbacks on extratropical storm tracks."

With respect to the second part of the comment: Rossby waves constitute undulations of the tropopause, i.e., northward and southward extensions of tropospheric and stratospheric air masses, respectively. By their very nature, Rossby waves thus comprise tropospheric and stratospheric PV anomalies. The reviewer is correct that relatively small heating rates in the stratosphere may yield relatively large cross isentropic transport of PV due to the large gradients in theta and PV. To the extent that the associated PV tendencies impact PV anomalies associated with Rossby waves, these tendencies are significant for the evolution of Rossby waves. Note however that we evaluate tendencies on an isentropic level intersecting the midaltitude tropopause (see Sect. 2.2, a few lines before Eq. 4).

6. Another comment on the radiation results would be that there has been significant focus on the effects of radiation, in particular related to cloud tops, on storm tracks (e.g., work by Aiko Voigt and the Cookie experiment community), and thus implicitly on cyclones on RWPs. The results of the authors indicate that the claimed impact by the aforementioned community might not be as relevant on a feature-based view and mainly reflect itself in a climatological background, which would be worthwhile to put in context.

This is another very good suggestion by the reviewer that will help to clarify the presentation of our results. Indeed, the results of our initially submitted manuscript indicated that radiative PV tendencies have - on average – a small impact on the life cycle of *individual* anomalies but that these tendencies project much more substantially on the background state. Considering

the spatial composites in the new Fig. 10 revises this interpretation: The local extrema in radiative PV tendencies are strongly connected to clouds and their magnitude is a substantial fraction of the "background" value. Our revised results thus indicate that cloud radiative effects may be important also for upper-level direct diabatic PV modification. We discuss our results now in the context of previous work that considered cloud radiative effects on storms tracks. It should be noted, however, that this effect is more complex than the upper-level direct diabatic PV tendencies considered herein and thus a comparison is not straightforward (for modified text see response to previous comment).

7. Related to the comment above, most likely, most heating associated with cloud and rain processes occurs below 600 hPa and therefore the tendencies of upper level PV are not directly affected by the diabatic terms. However, the displacement of the theta surfaces will be felt aloft, which will manifest itself in divergence at these levels. For example, heating at levels below 600 hPa yields a mass transport above potential temperature surfaces that can be located below 600 hPa initially, but the ensuing mass redistribution will reach higher altitudes in the hydrostatic and geostrophic adjustment. The divergent signal is thus potentially largely associated with the atmosphere trying to attain balance after experiencing diabatic heating. The method employed by the authors cannot disentangle between these causes and effects. The authors should expand the discussion about these potential caveats and what they imply for the interpretation of the results.

We are not sure if we understand the reviewer correctly. In particular, we do not see the different causes and effects that need to be disentangled. It might be helpful to re-state upfront that we evaluate our PV budget on isentropic levels, not on pressure levels. Pressure levels are used only in the technical sense of performing piecewise PV inversion.

The impact of diabatic heating due to mass transport to the upper-troposphere is fully accounted for in our analysis by the divergent term. We fully agree - and emphasize in the manuscript at several instances - that this is the major impact of latent heat release on RWPs. The divergent flow contains both, the "balanced" secondary circulation associated with latent heat release and unbalanced (gravity-bore like) motion. These two components are not disentangled by our diagnostic, but we believe that it is not these two components to that the reviewer refers. In addition, as discussed in some detail in the manuscript, it is difficult to disentangle the role of diabatic and adiabatic secondary circulations with our diagnostic, although first steps are being made towards this goal in the revised Sect. 5.3..

Latent heat release certainly generates PV anomalies below the maximum of heating. The balanced state associated with these anomalies, however, does not directly(!) impact the PV distribution on isentropic levels above. If the reviewer implied this impact by the wording "the ensuing mass redistribution will reach higher altitudes in the hydrostatic and geostrophic adjustment." the reviewer would not be correct. The PV distribution aloft is impacted by the diabatically generated lower-level PV anomalies indirectly(!) due to advection by the associated wind field. In L92 we explicitly state that we do not attempt to disentangle this impact, which has been the focus on many previous studies.

8. In general, I found it sometimes difficult to follow the reasoning of the authors and they sometimes also indicate that they will contradict themselves, e.g., paragraph 508-519, especially line 518. As a reader, I would appreciate a more stringent guidance through the material avoiding potential confusions and suspense to wait (at the cost that I might have forgotten until then) until further clarification in later sections.

Based on the reviewer's next comment, we will revise subsection 5.3 and thus the paragraph noted by the reviewer here explicitly. Otherwise, however, it is difficult for us to identify other parts in the manuscript that may require revisions based on this comment.

9. I find the argumentation in the ensuing paragraph also confusing (top of page 25). The authors make it sound like as if the moist baroclinic paradigm does not require a beneficial

phasing of the latent heating with the overall baroclinic structure. This would be incorrect and indeed the arguments presented by the authors at the end of the paragraph are consistent with the moist-baroclinic instability paradigm, i.e., the phase relation between heating and the perturbations is closely tied to the overall baroclinic structure. Furthermore, most of what is then argued in the last paragraph before the conclusions is straight forward moist-baroclinic instability reasoning, i.e., once there is a baroclinic growth, i.e., once a westward vertical tilt of the anomalies is established in a baroclinic environment, ascent and associated latent heating occurs at a favorable location for growth. The authors thus do not present something new in this context, even though they appear to make it seem like new. Instead of presenting these findings as something new, they should rather put their findings in context with existing literature on moist baroclinic instability and relate their findings to the overall three-dimensional structure of the RWP.

This is another good point of the reviewer. Moist-baroclinic instability theory links latent heat release to the region of dry ascent in a baroclinic wave. A beneficial phasing of latent heat release is thus inherent in the theory. Our original manuscript should have noted this relation more explicitly and have put our results in this context. The mere existence of beneficial phasing during moist baroclinic growth is certainly not a new result and our original manuscript was not sufficiently clear about the new aspects of our analysis. New aspects include considering data of a large number of real cases and going well beyond the modal structure of moist-baroclinic instability theory: we focus on the ridge amplification, which is much more pronounced than that of the remainder of the (putative) moist-baroclinic mode (e.g., the trough); we consider the variability of heating and its impact at similar stages of the baroclinic life cycle; and we investigate spatial patterns of (a proxy of) latent heating and its relation to PV anomalies that may evolve in time. The revised version of the manuscript now refers to previous literature of moist-baroclinic instability and will use this context to make clearer the new aspects of our study.

We have thoroughly revised the subsection. The first paragraph now clarifies and better motivates the purpose of the subsection. In addition, we have removed the claim that we present "a new qualitative hypothesis". The first paragraph reads: "This section addresses the question to what extent divergent ridge amplification can be attributed to the dry (balanced) dynamics of the baroclinically-growing wave and to latent heat release below. An answer to this question is of importance because it provides increased understanding of the sensitivities of the extent to which moist processes impact the large-scale flow by associated upper-tropospheric divergence. For the sake of brevity, we will abbreviate the divergent tendency, the baroclinic tendency, and the proxy for latent heat release in this subsection by DIV, BC, and LHRproxy, respectively. The basic idea is to use our data to attribute DIV to LHRproxy and to BC, respectively, with BC here serving as a simple proxy for the characteristics of the dry (balanced) dynamics, in particular for the state of the baroclinic development."

We have split the second paragraph into two. The first one discusses the strong coupling between dry dynamics and moist processes and concludes with the explicit statement that "This strong coupling between dry dynamics (BC) and moist processes (LHRproxy) during baroclinic growth has been noted in many previous studies and is consistent with the underlying assumptions of many moist-baroclinic instability theories (e.g., see references in introduction)."
The second one discusses the simple linear statistical analysis. Here we have somewhat revised the interpretation of this analysis, which now reads: "This simple statistical analysis confirms that ridge amplification by DIV is strongly coupled to the underlying baroclinic development, as signified here by BC. Variations of LHRproxy, when considered for all values of BC, are less well suited to describe linear variations in DIV. This simple perspective, however, should not be taken as evidence that upper-tropospheric divergence is predominantly due to secondary circulations associated with the balanced, dry dynamics of the growing baroclinic wave."

We now start the next paragraph with a clarification of its purpose: "Next, we consider in some more detail the variations of DIV with LHRproxy for different ranges of BC values."

The following description of the observations remain unchanged. Our interpretation of these observations, however, have undergone major revisions, following your justified critique. We first emphasize that "This result based on a large number of real-world cases is consistent with expectations from moist-baroclinic instability theories:" followed by describing the two salient aspects: "i) Figure 11c indicates the above-cited strong coupling between dry dynamics and moist processes during baroclinic growth. ii) Figure 11d indicates that the efficiency by which latent heat release leads to ridge building by divergent outflow depends on the underlying baroclinic development (BC)."

Before comparing the phasing aspect to extratropical transition, we now discuss this aspect in some more detail in the context of moist-baroclinic instabilty theories. This part serves to qualify our statement above that our results are consistent with these theories and should help to make clear that we do not claim to have found something fundamentally new here: "Our interpretation is that BC here contains this phasing information, i.e., information on the relative position of latent heat and the upper-level ridge. Few moist-baroclinic instability theories (Mak 1994, deVries et al. 2010) consider the phasing of latent heat release explicitly. More commonly, the strong coupling of moist processes to the dry dynamics inherent in the theories implies that latent heat release invigorates the ascent associated with dry dynamics (e.g., Emanuel et al. 1987) and thus latent heat release is most effective for moist-baroclinic growth when the phasing for dry baroclinic growth is most favorable."

We have revised the reference to extratropical transition to return from instability theories to the more specific case of divergent ridge building in real-world cases: "Specifically for ridge amplification by divergent outflow, the importance of favorable phasing has been emphasized in the context of the extratropical transition of tropical cyclones. In that context, the impact of latent heat release does not only depend on the magnitude of latent heat release but at least equally importantly on the relative position of latent heat release and the upper-tropospheric Rossby wave pattern (Keller et al. 2019, Riboldi et al 2019). Our examination of a large number of real-world cases indicates that this notion of favorable phasing transfers to the more general case of divergent ridge amplification within RWPs."

For the following paragraph we have clarified that the purpose is not only to examine the "phasing aspect" (eliminating the claim of a new hypothesis) but also "the relation of DIV with BC and LHRproxy". In addition, we have clarified the purpose of the design of the composites: "The design of these composites takes into account some of the variability of co-occurrence of LHR and BC, and thus moves beyond the strong coupling of moist processes and dry dynamics inherent in moist-baroclinic theories."

In the next paragraph, which describes the scenarios with near-median LHRproxy, we have added the explicit statement "Figures 12b,d,f thereby illustrate differences in the phasing of latent heat release and the upper-level ridge anomaly that occur on average at two different stages of baroclinic development in a large number of real-world cases." to link the discussion of the composites more strongly to the phasing discussion above." Besides a few minor changes of the wording , we have revised the end of the paragraph to conclude that "This striking coincidence strongly suggests that differences in divergent ridge amplification can be attributed to differences in the release of latent heat below in these scenarios also."

The last paragraph of the subsection has undergone major revisions. We now refrain from presenting "our phasing hypothesis" as a new result but rather focus on the relation of latent heat release and divergent ridge amplification: "Our analysis of the scenarios with near-median BC and large variations of LHRproxy and with near-median LHRproxy and large variations of BC (Figure 12) provides some evidence that divergent ridge building can predominantly be attributed to divergent outflow associated with latent heat release below.

The simple statistical analysis presented in Figure 11b may not provide similar evidence because that analysis does not account for changes in the pattern and phasing of latent heat release that occur on average during the baroclinic development." Our word of caution that concludes the paragraph remains unchanged.

Finally, we have revised statements in the abstract and conclusions accordingly. The second-last paragraphin the conclusions now reads (First sentence remains unchanged): "While there is evidence that the impact of latent heat release is most prominently communicated to RWPs by upper-tropospheric divergent outflow, it is in general difficult to accurately disentangle the relative contributions of dry and moist dynamics to upper-tropospheric divergence (e.g., Riemer et al., 2014; Quinting and Jones, 2016; Sanchez et al., 2020), and thus its impact on RWPs. This study does not attempt such an accurate quantitative decomposition. However, investigating scenarios with large variations of (a proxy for) latent heat release and baroclinic growth, respectively, while keeping the other process fixed at near-median values, does provide some evidence that divergent ridge building can predominantly be attributed to divergent outflow associated with latent heat release below. In addition, our results demonstrate that divergent ridge amplification does not only depend on the magnitude of latent heat release but also on the location of latent heat release relative to the upper-level PV anomalies. As expected from theories of moist-baroclinic instability (e.g., Emanuel et al. 1987, Mak 1994, deVries et a;. 2010), we observe that this phase relation becomes favorable when baroclinic growth commences and that it becomes increasingly more favorable when baroclinic growth increases. For real-world cases of divergent ridge amplification, the importance of phasing has first been noted explicitly in the context of extratropical transition (Keller et al. 2019, Riboldi et al. 2019). Our examination of a large number of real-world cases indicates that the notion of favorable phasing transfers to the more general case of divergent ridge amplification within RWPs."

The end of the abstract now reads: "Consistent with theories of moist-baroclinic instability, both the amplitude and the relative location of latent heat release within the developing wave pattern depends on the state of the baroclinic development. Taking this "phasing" aspect into account, we provide some evidence that variability in the strength of divergent ridge amplification can predominantly be attributed to variability in latent heat release below, rather than to secondary circulations associated with the dry dynamics of a baroclinic wave."

**Specific Comments:**
163: Has this approximate equality been checked with data? I am wondering how close this relation really holds.

Yes, it was tested with data. In the first authors PhD thesis, the advection of the background by the background flow has been investigated for different time mean durations and integrated over the whole inversion domain. It was shown that this term is negligible (2-3 orders of magnitude smaller) compared to the advection of background PV by the anomaly flow.

170-171: The authors state that the aforementioned occurrences are exceptional. Why do the authors not identify and quantify the potential influence of these occurrences and the effect mid-level PV anomalies might have on their results?

The impact of these occurrences might be interesting, but their analysis is beyond the scope of this manuscript. The focus of this study is the mean perspective of wave dynamics, not the impact of exceptional occurrences.

259: There is a grammar issue with the sentence, in particular with the run-on part "and

 mainly due to small-scale. . .". thanks

297-298: How can one infer the group velocity from the blue contours? Strictly speak–ing, given the 90 degrees shift, one can only identify the phase propagation.

This question is related to your third comment. We will make clearer in the revised version that amplification and decay of anomalies is defined by spatially integrated tendencies and will note explicitly that amplification and decay of anomalies occurs in this spatially integrated

sense. At the leading edge the quasi-barotropic tendencies generate and amplify new anomalies. At the trailing edge the quasi-barotropic tendencies lead to decay of anomalies. In that sense the quasi-barotropic PV tendencies relate to the group propagation of RWPs.

377: "save"? What does that mean in this context?

„save" in this context is used as a preposition (as a synonym for except)

399: The value is not "random", as the authors state it is reflecting a more "climatological" value.

Thank you, we agree that the term random is not a good choice and will rather use the term climatological in the revised version.

560: How can adiabatic subsidence be of significance in an isentropic framework? It would basically be invisible, except if it is associated with horizontal divergence.

We here refer to adiabatic subsidence in the context of our proxy for latent heat release. This proxy is calculated in pressure coordinates and thus reference to adiabatic subsidence does make sense.

614: "tropospheric"  thanks

637: Please clarify what is meant by "differences in phasing" in this context. Phasing of what? Furthermore, what aspects of the secondary circulation are meant?

"Differences in phasing" meant to refer to the relative position of latent heat release to upper-tropospheric PV anomalies. With "secondary circulations" we meant to refer to upper-tropospheric divergence associated with dry balanced dynamics. We have revised this statement (please see response to your last general comment), taking into further account revisions that we have implemented based on your last general comment above.

---

## Referee Report (RR1)

Second Review of WCD-2020-52:

"Potential-Vorticity Dynamics of Troughs and Ridges within Rossby Wave Packets
during a 40-year reanalysis period"
by
F. Teubler and M. Riemer

**General Comments:**

The authors have significantly reworked the manuscript based on the feedback they received. There are still some remarks that should be clarified.

Regarding my comment on the partitioning of PV anomalies, which the authors regard as "out-of-the-box" thinking, their response requires some clarification. In general, one uses interior PV and boundary conditions for potential temperature on the upper and lower boundaries as well as azimuthal wind conditions in the lateral direction to perform a complete PV inversion, assuming a geostrophic and hydrostatic state. These conditions can either be set to trivial values or be prescribed making certain assumptions. In my original comment, I wondered about the implications of the assumption that the PV anomaly in the respective domain would not be associated with a potential temperature anomaly on both vertical boundaries, which it appears is what the authors are assuming. While it might be common practice, it has also been shown that this can bear significant caveats (e.g., Egger and Spengler, 2018, DOI: 10.1175/JAS-D-17-0039.1).

In their response, the authors then blend arguments of regular PV inversion, including interior PV and boundary conditions as outlined above, with PV thinking of delta sheets introduced by Bretherton (1966), where all information is condensed into delta perturbations of potential temperature, i.e., static stability anomalies, at the boundaries, which can be extended to a given amount of these discrete delta layers of these type of PV anomalies (e.g., Heifetz et al., 2004, de Vries et al., 2009). Based on my reading of the manuscript, however, the PV inversion outlined by the authors deals with interior PV in given volumes and potential temperature anomalies on the vertical boundaries, which is also what my comment and question was based on. As indicated previously, the exact inversion procedure including the assumptions should be clearly outlined and a justification of the assumptions, especially regarding the boundary conditions, would be highly appreciated.

My comment was not about the respective PV anomaly projecting on static stability anomalies, but on the used potential temperature anomalies in the inversion, as it is these anomalies, based on my understanding of the authors' method, that are used for the inversion. Regarding my comment, it would be correct that these potential temperature anomalies are then part of the respective PV anomaly. In their response, the authors then use an argument that would refute their own assumption, where they argue that vertical velocity is negligible at the upper and lower boundary, thereby suppressing an influence of the distant PV anomaly on this surface. However, the same argument would then also apply to the PV in proximity of the boundary if it is seen as more or less rigid. Therefore, no potential temperature anomalies along the boundaries would be attributable to any PV anomalies, though one appears to happily make certain a priori choices.

A thought experiment; if one wants to infer the anomalous flow field in the lower domain based on the PV in the upper domain, actually no information of the PV in the upper domain is needed if one knows the potential temperature at the interface between the

upper and lower domain as well as the potential temperature at the surface, given that one inverts zero PV in the lower domain. Thus, the main question would be how the PV of the upper domain would project on the potential temperature at these interfaces. If a priori prescribing half of that information, what is one really assessing? To further substantiate the claims of the authors, sensitivity experiments could be performed by moving the vertical boundaries of the chosen pressure levels as well as varying the choices of the boundary conditions. If the interior PV really is the most significant detail, then the respective results should not depend significantly on these choices if the volumes are chosen as such that the PV that is regarded as significant is always contained in the chosen volumes. In addition, a completeness test could be performed to further substantiate the assumptions of the authors with respect to their choices, where one would check if the sum of the anomalies yields the total field, or how closely it would resemble it. If the total field cannot be satisfactorily reconstructed, this would also point to deficiencies of the method and/or choices for the inversion.

There was a conflict in the response of the authors, where they in response to my comment 7. clearly state that they are arguing in an isentropic framework, though in response to my comment on line 560 they refer to a calculation and argument in pressure coordinates. The implications of this shift of framework in argumentation should be clarified. As indicated, adiabatic displacements would be invisible in an isentropic framework, except for implied divergence between isentropic surfaces. As the authors now include a more detailed discussion of the divergence, this appears to be more directly linked, but the authors are encouraged to double check for potential inconsistencies.

Regarding the authors' response to my point 7., they claim that my comment contained an incorrect statement. This is not true and in a way the reviewers more or less merely paraphrased my comment. To further clarify, with "ensuing", an effect, which can be referred to as indirect, of the heating was implied in my comment, i.e., the heating yields a secondary circulation redistributing mass and vorticity and thereby PV. It appears the authors are aware of that. Furthermore, the more detailed discussion of the role of the divergence relates to my previous comment and thus addresses some of my concerns, in particular the relation of the divergence to the diabatic heating.

---

## Author Response (AR2)

**Reply to Reviewer 2**

General Comments:

The authors have significantly reworked the manuscript based on the feedback they received. There are still some remarks that should be clarified.

We are glad to see that our revisions clarified many of the concerns of the first review of this reviewer. We appreciate the reviewer's additional comments that will help to further clarify the presentation of aspects of the methods applied in our study.

A summary of the specific changes made in the revised manuscript are given at the end of this response document.

Regarding my comment on the partitioning of PV anomalies, which the authors regard as "out-of-the-box" thinking, their response requires some clarification. In general, one uses interior PV and boundary conditions for potential temperature on the upper and lower boundaries as well as azimuthal wind conditions in the lateral direction to perform a complete PV inversion, assuming a geostrophic and hydrostatic state. These conditions can either be set to trivial values or be prescribed making certain assumptions. In my original comment, I wondered about the implications of the assumption that the PV anomaly in the respective domain would not be associated with a potential temperature anomaly on both vertical boundaries, which it appears is what the authors are assuming. While it might be common practice, it has also been shown that this can bear significant caveats (e.g., Egger and Spengler, 2018, DOI: 10.1175/JAS-D-17-0039.1).

In their response, the authors then blend arguments of regular PV inversion, including interior PV and boundary conditions as outlined above, with PV thinking of delta sheets introduced by Bretherton (1966), where all information is condensed into delta perturbations of potential temperature, i.e., static stability anomalies, at the boundaries, which can be extended to a given amount of these discrete delta layers of these type of PV anomalies (e.g., Heifetz et al., 2004, de Vries et al., 2009). Based on my reading of the manuscript, however, the PV inversion outlined by the authors deals with interior PV in given volumes and potential temperature anomalies on the vertical boundaries, which is also what my comment and question was based on. As indicated previously, the exact inversion procedure including the assumptions should be clearly outlined and a justification of the assumptions, especially regarding the boundary conditions, would be highly appreciated.

We agree with the reviewer that inversion of interior PV in a limited domain is (in a mathematical sense) not a well-posed problem because the (lateral and vertical) boundary conditions that are associated with the balanced state, i.e., the solution of the inversion, are not known a priori. Further issues arise when performing piecewise inversion using nonlinear balance because of the nonlinearity of the balance condition. We had appreciated the mathematical inexactness of PV inversion in the original version of our manuscript at the end of section 2.2. We agree with the reviewer that our list of issues mentioned at that point is incomplete and the list in the revised version now includes imperfect knowledge of the vertical boundaries.

We further agree that the associated discussion was too brief. The revised version now includes a much-extended discussion on the vertical boundaries and a quantitative estimate of uncertainty associated with piecewise PV inversion (see below).

My comment was not about the respective PV anomaly projecting on static stability anomalies, but on the used potential temperature anomalies in the inversion, as it is these anomalies, based on my understanding of the authors' method, that are used for the inversion. Regarding my comment, it would be correct that these potential temperature anomalies are then part of the respective PV anomaly. In their response, the authors then use an argument that would refute their own assumption, where they argue that vertical velocity is negligible at the upper and lower boundary, thereby suppressing an influence of the distant PV anomaly on this surface. However, the same argument would then also apply to the PV in proximity of the boundary if it is seen as more or less rigid. Therefore, no potential temperature anomalies along the boundaries would be attributable to any PV anomalies, though one appears to happily make certain a priori choices.

If the vertical boundaries were indeed rigid, i.e., w=0 at these boundaries, interior and boundary anomalies could be clearly distinguished. (Anomalies on the boundaries can of course still exist due to advection of (boundary) potential temperature across an existing background gradient, as in the Eady model.) The distinction starts to blur when the boundaries are moved into the interior and w=0 no longer holds. In our current study, however, we do not distinguish between the lower boundary theta anomalies and the low-level interior PV anomalies (analogously for the upper levels). In our previous response to this reviewer, we had thus argued only why it is a very good assumption that the impact of the distant anomalies on boundary theta anomalies is small.

Defining the low-level PV anomaly (in the generalized sense of Bretherton (1966)) as the combination of the lower boundary theta anomalies and the low-level interior PV anomalies makes physical sense because inversion of these anomalies considers, e.g., for a cyclone the impact of the thermal wave and the diabatically generated interior anomalies in combination. Other studies may seek to distinguish between these anomalies, but this is not the goal of the current study. Due to the combination of both anomalies, there is no need to identify the theta anomaly at the lower boundary that is associated with the low-level interior PV anomaly.

The lower boundary is physically motivated, due to the existence of the Earth's surface. Moving this boundary into the interior, i.e., at the top of the boundary layer, is physically motivated, too, because one attempts to avoid boundary layer effects, e.g., the diurnal cycle of theta anomalies.

The situation is different at upper levels because there is no boundary in a physical sense. The vertical boundary there needs to be introduced when seeking a numerical solution to the inversion problem with available data. The upper-level boundary is situated in the lower-most stratosphere and thus intersects the PV anomalies associated with RWPs. To a large extent, theta anomalies at this boundary can thus be assumed to be due to the stability anomaly associated with such PV anomalies in the lower-most stratosphere. PV anomalies higher up in the stratosphere that are unrelated to RWPs (as well as those at low levels; see response to first review) may contribute to theta anomalies at that boundary by their far field effect (''action-at-a-distance''). This effect, however, depends on static stability, which is very high in the stratosphere, and the effect can thus be expected to be negligibly small. Overall, we consider our interpretation of the boundary theta anomalies as lower and upper-level PV anomalies, respectively, to be an excellent approximation (for the reasons given here and in our reply to the first review of this reviewer).

The revised version of the manuscript reflects this discussion when introducing the definition of PV anomalies in section 2.2.

A thought experiment: if one wants to infer the anomalous flow field in the lower domain base on the PV in the upper domain, actually no information of the PV in the upper domain is needed if one knows the potential temperature at the interface between the upper and lower domain as well as the potential temperature at the surface, given that one inverts zero PV in the lower domain. Thus, the main question would be how the PV of the upper domain would project on the potential temperature at these interfaces. If a priori prescribing half of that information, what is one really assessing?

We fully agree with the reviewer's reservation about the boundary condition in this thought experiment. The thought experiment nicely illustrates that the boundaries of an elliptic problem are an integral part and need to be chosen in accordance with the physical problem at hand. The thought experiment is applicable to a boundary that is introduced at say 500 hPa. However, as described above, the situation is qualitatively very different in our study.

To further substantiate the claims of the authors, sensitivity experiments could be performed by moving the vertical boundaries of the chosen pressure levels as well as varying the choices of the boundary conditions. If the interior PV really is the most significant detail, then the respective results should not depend significantly on these choices if the volumes are chosen as such that the PV that is regarded as significant is always contained in the chosen volumes. In addition, a completeness test could be performed to further substantiate the assumptions of the authors with respect to their choices, where one would check if the sum of the anomalies yields the total field, or how closely it would resemble it. If the total field cannot be satisfactorily reconstructed, this would also point to deficiencies of the method and/or choices for the inversion.

In our previous studies using piecewise PV inversion we had performed sensitivity tests and found little sensitivity of the results to the specific height of the vertical boundaries (within a physically reasonable range, namely 175hPa and 75 hPa at upper levels and 825 hPa at lower levels). We thus feel very comfortable using the setup from previous studies here. We had performed also extensive tests of the reconstruction of the upper-level winds. Errors relative to the total wind are on average very small (below 5 %). This test is somewhat incomplete, though, because the ''baroclinic'' wind component at upper levels is relatively small, too, such that small relative errors of the total wind may still affect the baroclinic interaction. In fact, we thus did include a completeness test graphically in the original manuscript with respect to the contribution of the baroclinic and quasi-barotropic wind components to amplitude evolution (shading in Fig. 7). Thanks to the reviewer, we now realize our oversight that we failed to describe this assessment of the uncertainty associated with piecewise PV inversion in the main text; we had mentioned it in passing only in the caption. On average, the relative uncertainty is small. For troughs during extended winter (Fig. 7b), however, there is consistently a relative uncertainty associated with piecewise PV inversion of 25% - 30% in the baroclinic component, if the uncertainty is split equally between the baroclinic and the quasi-barotropic contribution. Even if the uncertainty were attributed completely to the baroclinic component, however, none of the results of our study would be affected. The revised version of the manuscript now includes this description of the relative uncertainty associated with piecewise PV inversion.

There was a conflict in the response of the authors, where they in response to my comment 7. clearly state that they are arguing in an isentropic framework, though in response to my comment on line 560 they refer to a calculation and argument in pressure coordinates. The

implications of this shift of framework in argumentation should be clarified. As indicated, adiabatic displacements would be invisible in an isentropic framework, except for implied divergence between isentropic surfaces. As the authors now include a more detailed discussion of the divergence, this appears to be more directly linked, but the authors are encouraged to double check for potential inconsistencies.

We realize that our presentation of the proxy for latent heat release leaves room for misinterpretation. We do not use this proxy to calculate associated diabatic PV tendencies. Rather, we use the proxy to explore the relation of upper-tropospheric divergence with latent heat release below, i.e., in the lower to mid-troposphere. For the definition of ''lower to mid-troposphere'' we use the pressure levels given in the text (1000 hPa - 500 hPa). The revised version now emphasizes more clearly the purpose of our proxy for latent heat release.

Regarding the authors' response to my point 7., they claim that my comment contained an incorrect statement. This is not true and in a way the reviewers more or less merely paraphrased my comment. To further clarify, with "ensuing", an effect, which can be referred to as indirect, of the heating was implied in my comment, i.e., the heating yields a secondary circulation redistributing mass and vorticity and thereby PV. It appears the authors are aware of that. Furthermore, the more detailed discussion of the role of the divergence relates to my previous comment and thus addresses some of my concerns, in particular the relation of the divergence to the diabatic heating.

We are glad to see that the extended discussion of the role of divergence is helpful to address this previous comment of the reviewer.

**Specific changes to the manuscript:**

Instead of simply writing in Sect. 2.2.: "PV anomalies are partitioned into upper-level PV anomalies including the upper-boundary θ-anomalies and into lower-level PV anomalies including the lower-boundary θ-anomalies." we have now added a new paragraph that discusses the interpretation of theta anomalies in some depth: "The upper and lower boundary $\theta$ anomalies are included in our definition of the upper- and lower-level PV anomalies, respectively. The lower boundary $\theta$ anomalies include the thermal anomalies of baroclinic waves, e.g., warm and cold sectors of cyclones, and a contribution from the static stability anomalies associated with low-level (interior) PV anomalies. Our upper boundary is situated in the lower-most stratosphere intersecting PV anomalies that are associated with RWPs. The upper boundary $\theta$ anomalies thus include the static stability anomalies associated with these PV anomalies. In addition, the boundary $\theta$ anomalies may include contributions from distant PV anomalies. Namely, the lower boundary $\theta$ anomaly may include contributions from PV anomalies above the separation level and the upper boundary $\theta$ anomaly contributions from PV anomalies below the separation level and from stratospheric PV anomalies outside of our domain that are unrelated to RWPs. These contributions are ultimately due to vertical motion associated with the evolution of the distant anomalies (cf.\, the concept of a "very gentle 'vacuum cleaner'" in Sect.\, 4 of \citet{HoskinsEtAl1985}). Vertical motion at the upper and lower boundary, however, is strongly limited by the large static stability in the stratosphere and the closeness of the boundary to the rigid boundary of the Earth's surface, respectively. Contributions of distant PV anomalies can thus be expected to be negligibly small, and we consider our interpretation

of the boundary $\theta$ anomalies as upper-and lower-level PV anomalies to be an excellent approximation."

Having given the vertical boundaries much attention, we have simply deleted "lateral" before "boundaries" when we list the sources of inexactness at the end of Sect. 2.2.

At the end of Sect. 2.2, we have added the following discussion of uncertainty in the results of our piecewise PV inversion: "Figure 7 shows how these inherent and numerical inaccuracies of our piecewise PV inversion affect the results. On average, the relative uncertainty is small. For troughs during extended winter (Figure7b), however, there is a persistent relative uncertainty of 25% - 30% in the baroclinic component, when the uncertainty is split equally between the baroclinic and the quasi-barotropic tendency. Despite this relatively large uncertainty, and even if all of the uncertainty were attributed to the baroclinic component, none of the results of our study would be qualitatively affected."

Introducing the proxy for latent heat release when discussing Fig. 3, we now write: "… consistent with the invigoration of ascent and thus upper-tropospheric divergence by latent heat release below*, i.e., in the lower to mid-troposphere*. To examine the role of *lower to mid-tropospheric* latent heat release, we consider as a proxy …" (additions in italics)
Furthermore, we have added "in the lower to mid-troposphere" to the caption of Fig. 3.

---

## Author Response (AR3)

**Reply to Reviewer #2**

The authors expanded the discussion around the challenges of PV inversion and included valuable information on the sensitivity of their choices, which is highly appreciated and will certainly help the reader to better understand the method and its potential limitations.

Overall, the response and changes are satisfactory, though I would like to ask the authors to refrain from using "excellent" in line 191 (line ref. track changes version). As they point out themselves, there are several issues with the choices for PV inversion, so "reasonable" of "justifiable" appears to be a more appropriate wording.

We are glad to read that the reviewer considers our revisions to be helpful and our response and changes overall satisfactory. We agree that „excellent" is too strong of a wording, in particular because we are not able to quantify the smallness of the degree to which distant anomalies affect the interpretation of boundary theta anomalies as PV anomalies attributable to the upper- and lower level components of baroclinic RWPs.
We would like to point out, though, that „excellent" does not refer to "several issues with the choices for PV inversion" but at this point specifically to the interpretation of the upper- and lower-boundary theta anomalies. Our rather detailed discussions in the first two responses provide our rationale why we are confident that our interpretation implies a very accurate approximation. We still agree that softening our statement at this point is appropriate. Adopting wording suggested by the reviewer, we have changed the second part of the sentence to „…, and the interpretation of the boundary theta-anomalies as upper-and lower-level PV anomalies appears to be very reasonable."